# Gamma distribution model of diffusion MRI for the differentiation of primary central nerve system lymphomas and glioblastomas

Osamu Togao[1]*, Toru Chikui[2], Kenji Tokumori[3], Yukiko Kami[2], Kazufumi Kikuchi[4], Daichi Momosaka[4], Yoshitomo Kikuchi[4], Daisuke Kuga[5], Nobuhiro Hata[5], Masahiro Mizoguchi[5], Koji Iihara[5], Akio Hiwatashi[4]

1 Department of Molecular Imaging & Diagnosis, Graduate School of Medical Sciences, Kyushu University, Fukuoka, Japan, 2 Department of Oral and Maxillofacial Radiology, Faculty of Dental Science, Kyushu University, Fukuoka, Japan, 3 Department of Clinical Radiology, Faculty of Medical Technology, Teikyo University, Fukuoka, Japan, 4 Department of Clinical Radiology, Graduate School of Medical Sciences, Kyushu University, Fukuoka, Japan, 5 Department of Neurosurgery, Graduate School of Medical Sciences, Kyushu University, Fukuoka, Japan

* togao@radiol.med.kyushu-u.ac.jp

**Data Availability Statement:** All relevant data are within the manuscript and Supporting Information file.

## Abstract

The preoperative imaging-based differentiation of primary central nervous system lymphomas (PCNSLs) and glioblastomas (GBs) is of high importance since the therapeutic strategies differ substantially between these tumors. In this study, we investigate whether the gamma distribution (GD) model is useful in this differentiation of PNCSLs and GBs. Twenty-seven patients with PCNSLs and 57 patients with GBs were imaged with diffusion-weighted imaging using 13 b-values ranging from 0 to 1000 sec/mm². The shape parameter ($\kappa$) and scale parameter ($\theta$) were obtained with the GD model. Fractions of three different areas under the probability density function curve (f1, f2, f3) were defined as follows: f1, diffusion coefficient (D) $<1.0\times10^{-3}$ mm²/sec; f2, D $>1.0\times10^{-3}$ and $<3.0\times10^{-3}$ mm²/sec; f3, D $>3.0\times10^{-3}$ mm²/sec. The GD model-derived parameters were compared between PCNSLs and GBs. Receiver operating characteristic (ROC) curve analyses were performed to assess diagnostic performance. The correlations with intravoxel incoherent motion (IVIM)-derived parameters were evaluated. The PCNSL group's $\kappa$ (2.26 ± 1.00) was significantly smaller than the GB group's (3.62 ± 2.01, p = 0.0004). The PCNSL group's f1 (0.542 ± 0.107) was significantly larger than the GB group's (0.348 ± 0.132, p<0.0001). The PCNSL group's f2 (0.372 ± 0.098) was significantly smaller than the GB group's (0.508 ± 0.127, p<0.0001). The PCNSL group's f3 (0.086 ± 0.043) was significantly smaller than the GB group's (0.144 ± 0.062, p<0.0001). The combination of $\kappa$, f1, and f3 showed excellent diagnostic performance (area under the curve, 0.909). The f1 had an almost perfect inverse correlation with D. The f2 and f3 had very strong positive correlations with D and f, respectively. The GD model is useful for the differentiation of GBs and PCNSLs.

**Funding:** This work was supported by JSPS KAKENHI Grant Number JP17K10410 and JP20K08111. O.T. recieved these grants. https://www.jsps.go.jp/j-grantsinaid/ The funders had no role in study design, data collection and analysis, decision to publish, or preparation of the manuscript.

**Competing interests:** The authors have declared that no competing interests exist.

## Introduction

The preoperative imaging-based differentiation of primary central nervous system lymphomas (PCNSLs) and glioblastomas (GBs) is of high importance since the therapeutic strategies differ substantially between these tumors [1, 2]. The treatment of GBs is based on the maximal possible safe surgical resection together with postoperative chemoradiation therapy [1] whereas PCNSLs require a biopsy for histological confirmation followed by chemoradiation therapy [2]. In typical cases, the differentiation of these tumors by conventional MRI is not always difficult since PCNSLs shows homogenous contrast enhancing lesions while GBs show irregular and heterogenous ring enhancing lesion with necrosis. However, it is frequently difficult to differentiate these tumors especially when they demonstrate atypical imaging features.

Several studies have indicated that advanced MRI techniques such as diffusion-weighted imaging (DWI) [3–6], dynamic susceptibility contrast perfusion-weighted imaging [6–8], and arterial spin labeling [9] are useful for distinguishing PCNSLs and GBs. According to those studies, PCNSLs are characterized by more restricted water diffusion and lower perfusion compared to GBs.

Many mathematical models have been proposed for the analysis of diffusion MRI. The mono-exponential model describes the Brownian motion of water molecules by calculating the apparent diffusion coefficient (ADC) based on the Gaussian distribution of diffusion displacement [3]. The bi-exponential intravoxel incoherent motion (IVIM) model aims to separate the true water diffusion and the capillary perfusion by using multiple low b-values [10, 11]. Diffusion kurtosis imaging (DKI) is an approach used to characterize non-Gaussian water diffusion, which estimates kurtosis metrics [12].

It has been reported that all of these approaches are useful in differentiating GBs and PCNSLs [3, 13, 14], but all have possible limitations. The mono-exponential model may not precisely reflect the reality of diffusion behavior in heterogenous biological tissues, since this model assumes a Gaussian distribution. The bi-exponential model could be influenced by an uncertainty of the estimated perfusion, since signal measurements at low b-values are susceptible to measurement errors [15–18]. The DKI model is limited by the unclear biological interpretation of the kurtosis parameters [18–21].

As one of the non-Gaussian distribution models, a statistical model based on the gamma distribution (GD) has been proposed for diffusion MRI analyses [22]. The GD model is a two-parameter family of continuous probability distribution parametrized in terms of the shape parameters kappa ($\kappa$) and the scale parameter theta ($\theta$), and this model assumes that the diffusion coefficient (D) is distributed continuously within a voxel. The GD model allows us to estimate fractions of a tissue type based on the concept that the area fractions for $D < 1.0 \times 10^{-3}$ mm$^2$/sec, $D = 1.0 \times 10^{-3}$ to $3.0 \times 10^{-3}$ mm$^2$/sec, and $D > 3.0 \times 10^{-3}$ mm$^2$/sec are attributed to intracellular, extracellular extravascular, and intravascular spaces, respectively [18, 22, 23]. Based on these fractions, we may be able to estimate histopathological conditions of neoplasms or organs.

The GD model has been used to assess prostate cancers [22–24], breast cancers [18], and renal function [25]. The GD model was also used to assess cerebral ischemic stroke in rat brains, and it was showed that this model exhibited a better performance than the conventional mono-exponential model and allowed for a significantly enhanced visualization of ischemic lesions [26]. To the best of our knowledge, its application to brain tumors has never been reported. We conducted the present study to determine whether the GD model is useful in the differentiation of PCNSLs and GBs.

## Materials and methods

This retrospective study was approved by the Institutional Review Board of Kyushu University Hospital (no. 2019–447), and the requirement for informed consent was waived.

## Patients

The DWI protocol with multiple b-values has been a part of our routine preoperative MRI examination for patients with brain tumors since January 2013. The patient inclusion criteria for this study were: (1) The DWI with multiple b-values was conducted preoperatively for the patient during the period from January 2013 to August 2019; and (2) The patient subsequently underwent a surgical resection or biopsy within 1 month of the DWI with multiple b-values, and the histopathological diagnosis of PCNSL or GB was made. A total of 89 patients met these criteria. The exclusion criteria were as follows: (1) no distinct contrast enhancement observed in the lesion (n = 3); and (2) difficulty in the analysis of images due to severe artifacts (n = 2). Thus, a total of 84 patients including 27 with PCNSLs (age, 62.9 ± 15.5 years; Male, 17 patients; Female, 10 patients) and 57 with GBs (age, 66.0 ± 16.4 years; Male, 31 patients; Female, 26 patients) were included in this study. The difference between the number of patients with PCNSLs and GBs can be explained by the fact that the PCNSLs are less frequent compared to the GBs [27].

## MRI

Multi-b-value DWI was performed on a 3T clinical scanner (Achieva 3.0TX or Ingenia 3.0T, Philips Healthcare, Best, The Netherlands) with an 8-channel or 15-channel head coil. The DWI was performed in axial planes by using a single-shot echo-planar imaging diffusion sequence, with 13 b-values (0, 10, 20, 30, 50, 80, 100, 200, 300, 400, 600, 800, 1000 sec/mm$^2$) in three orthogonal directions. The other imaging parameters were: repetition time, 2,500 msec; echo time, 70 msec; matrix, 128×126 (reconstructed to 256×256); slice thickness, 5 mm, field of view, 230×230 mm; number of slices, 11; sensitivity encoding factor, 1.5; scan time, 2 min 7 sec. For reference, several standard MR images including contrast-enhanced T1-weighted images were acquired.

## Image analysis

The mono-exponential model was computed using all of the above-listed b-values according to the following equation:

$$\frac{S_b}{S_0} = e^{-b \times ADC} \tag{1}$$

where Sb is the signal intensity for each b-value and S0 is the signal intensity at a b-value of zero.

In the bi-exponential model, the signal decay was estimated by the following the equation:

$$\frac{S_b}{S_0} = (1-f) \cdot e^{-bD} + f \cdot e^{-bD*} \tag{2}$$

where D* is the pseudo-diffusion coefficient, and the f is the volume fraction within a voxel of water flowing in perfused capillaries.

The GD model is represented by ρ(D) and is given by:

$$\rho(D) = \frac{1}{\Gamma(\kappa)\theta^{\kappa}} \cdot D^{\kappa-1} \cdot exp\left(\frac{-D}{\theta}\right) \tag{3}$$

where κ describes the shape parameter and θ describes the scale parameter. When the distribution of D follows this equation, the signal intensity on DWI is given by:

$$S(b) = S0 \cdot \frac{1}{(1+\theta b)^{\kappa}} \tag{4}$$

Three different areas under the probability density function (PDF) curve were defined as follows: f1, the fraction of D $<1.0\times10^{-3}$ mm$^2$/sec; f2, the fraction of $1.0\times10^{-3}$ to $3.0\times10^{-3}$ mm$^2$/sec; f3, the fraction of D $>3.0\times10^{-3}$ mm$^2$/sec. The f1 value is attributed to the intracellular component; the f2 is attributed to the extracellular extravascular component, and the f3 is attributed to the intravascular component [18, 22, 23].

The DWI data in the digital imaging and communications in medicine (DICOM) format were transferred to a personal computer and fit to the GD model, and then the κ and θ values were estimated using the Image J software program (ver. 1.52p; U.S. National Institutes of Health, Bethesda, MD) and self-built plug-ins. After the export of the x- and y-coordinates and the κ and θ of each pixel within the region of interest (ROI), the f1, f2, and f3 values of each pixel were calculated using Microsoft Excel ver. 16.16.14.

## ROI placement

The matrix sizes of the postcontrast T1-weighted images were adjusted to the same size as those of the DWI using the ImageJ function to match the geometric information of these images. ROIs were placed to delineate the enhancing lesion on the single slice that had the maximum area. On the size-adjusted postcontrast T1-weighted images, areas showing contrast enhancement were manually segmented by a neuroradiologist with 19 years of experience (O. T.) (Fig 1). The areas with necrosis, cystic lesion, hemorrhage, or obvious artifacts were carefully excluded from the ROI.

The ROIs were copied from the postcontrast T1-weighted images and pasted to the DWI. Fine manual adjustments were made when there were locational mismatches due to image distortion or the patient's motion, etc. The ROIs were also placed on the peritumoral non-contrast-enhancing T2-hyperintense areas to evaluate whether there were differences in histological features including tumor infiltration or increased vascularity in the peritumoral areas between PCNSLs and GBs. In addition, the ROIs were placed on the contralateral normal-appearing white matter. The ROIs for the peritumoral non-contrast-enhancing T2-hyperintense areas and contralateral normal-appearing white matter were measured on the image obtained with the b-value of 0 sec/mm$^2$ image. The same ROIs were used for all DWI analyses.

## Statistical analyses

The GD model-derived and IVIM-derived parameters were compared between the PCNSLs and GBs with the Mann-Whitney U-test. A receiver operating characteristic (ROC) curve analysis was performed to assess the diagnostic performance of the parameters in the differentiation of PCNSLs and GBs. The area under the curve (AUC) was calculated, and then the sensitivity and specificity were obtained. The optimal cutoff point was determined by Youden's method [28]. The diagnostic performance was considered excellent for AUC values between 0.9 and 1.0, good for AUC values between 0.8 and 0.9, fair for AUC values between 0.7 and 0.8, poor for AUC values between 0.6 and 0.7, and failed for AUC values between 0.5 and 0.6 [29].

To determine whether the combination of multiple parameters for both the GD model and the IVIM model increases the diagnostic performance, we first performed a stepwise analysis to select the explanatory variables for a multiple regression model from a group of candidate variables by going through a series of automated steps. A forward-selection rule was applied in which the analysis started with no explanatory variables and then added variables, one by one, based on which variable was the most statistically significant, until there were no remaining statistically significant variables [30, 31]. We then performed a binomial logistic regression analysis to examine the AUCs of the combinations of the selected parameters. Two

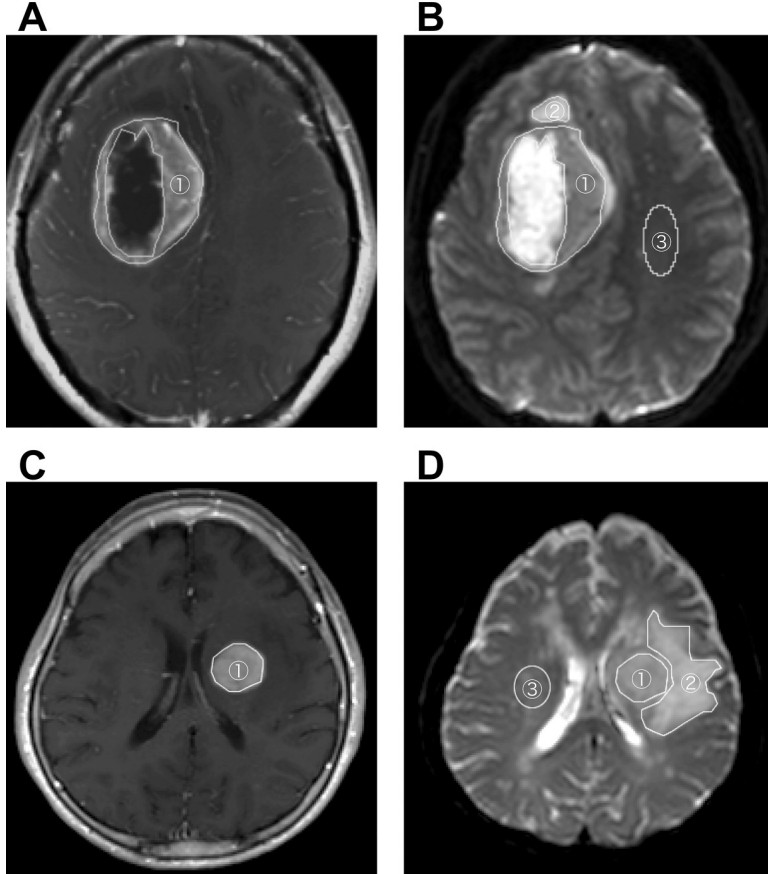

**Fig 1. Regions-of-interest (ROIs).** Fig 1A and 1B show a GB with ring enhancement, and Figures C and D show a PCNSL with solid enhancement. The ROIs were placed on postcontrast T1-weighted images to include contrast enhancing lesions (**A**, **C**, area #1). The ROIs were also placed on the non-contrast-enhancing T2-hyperintense areas surrounding the contrast-enhancing area (area #2) and the contralateral normal-appearing white matter (**B**, **D**, area #3).

independent AUCs were compared using the method of Delong et al. [32]. The correlations among the parameters were assessed with Pearson' correlation. Statistical analyses were performed with Prism 5.0 (GraphPad Software, San Diego, CA), MedCalc 19.1 (Broekstraat, Mariakerke, Belgium), and JMP Pro 14.0 (SAS Institute, Cary, NC). P-values <0.05 were considered significant.

## Results

### Comparisons of the parameters between the PCNSL and GB groups

The detailed information for the parameters in the gadolinium enhancing lesion, peritumoral T2-hyperintense areas without contrast enhancement, and normal appearing white matter is summarized in **Table 1**.

The results of our comparisons of the GD model-derived parameters between the PCNSLs and GBs in the gadolinium-enhancing lesions are shown in **Fig 2**. In the gadolinium-enhancing lesions, the κ was significantly smaller in the PCNSL group (2.26 ± 1.00) than in the GB group (3.62 ± 2.01, p = 0.0004), the f1 was significantly larger in the PCNSL group (0.542 ± 0.107) than in the GB group (0.348 ± 0.132, p<0.0001), the f2 was significantly smaller

**Table 1. Gamma distribution model-derived parameters in PCNSLs and GBs.**

| | | κ | | θ (×10⁻⁶ mm²/s) | | f1 | | f2 | | f3 | |
|---|---|---|---|---|---|---|---|---|---|---|---|
| **Enhancing lesion** | PCNSL | 2.26±1.00 | p = 0.0004 | 1.91±2.43 | p = 0.6341 | 0.542±0.107 | p<0.0001 | 0.372±0.098 | p<0.0001 | 0.086±0.043 | p<0.0001 |
| | GB | 3.62±2.01 | | 1.72±1.63 | | 0.348±0.132 | | 0.508±0.127 | | 0.144±0.062 | |
| **T2-hyperintense areas** | PCNSL | 8.23±2.42 | p = 0.8528 | 0.39±0.19 | p = 0.4747 | 0.140±0.066 | p = 0.4014 | 0.775±0.074 | p = 0.8882 | 0.085±0.045 | p = 0.2570 |
| | GB | 8.36±3.18 | | 0.40±0.25 | | 0.188±0.150 | | 0.752±0.128 | | 0.073±0.040 | |
| **NAWM** | PCNSL | 2.76±1.18 | p = 0.6341 | 0.85±0.45 | p = 0.1436 | 0.642±0.047 | P<0.0001 | 0.316±0.036 | p<0.0001 | 0.043±0.026 | p = 0.0105 |
| | GB | 2.55±0.75 | | 1.08±1.41 | | 0.593±0.044 | | 0.354±0.038 | | 0.053±0.019 | |

PCNSL, primary central nerve system lymphoma; GB, glioblastoma; NAWM, normal appearing white matter.

in the PCNSL group (0.372 ± 0.098) than in the GB group (0.508 ± 0.127, p<0.0001), and the f3 was significantly smaller in the PCNSL group (0.086 ± 0.043) than in the GB group (0.144 ± 0.062, p<0.0001), while the θ was not significantly different between the groups.

In the peritumoral T2-hyperintense areas without contrast enhancement, no significant differences were found between the PCNSL and GB groups for any of the GD model derived parameters.

In the contralateral normal-appearing white matter, the f1 was significantly larger in the PCNSL group (0.642 ± 0.047) than in the GB group (0.593 ± 0.044, p<0.0001), the f2 was significantly smaller in the PCNSL group (0.316 ± 0.036) than in the GB group (0.354 ± 0.038, p<0.0001), and the f3 was significantly smaller in the PCNSL group (0.043 ± 0.026) than in the GB group (0.053 ± 0.019, p = 0.0105).

**Fig 3** provides a PCNSL case that showed a ring-like enhancing mass lesion mimicking a GB. This lesion showed a low κ, a large f1, a small f2, and a small f3, suggesting PCNSL. **Fig 4** demonstrates a GB case that showed a solid enhancing mass lesion. This lesion showed a small κ, a small f1, moderate f2 and large f3, which are consistent with GB.

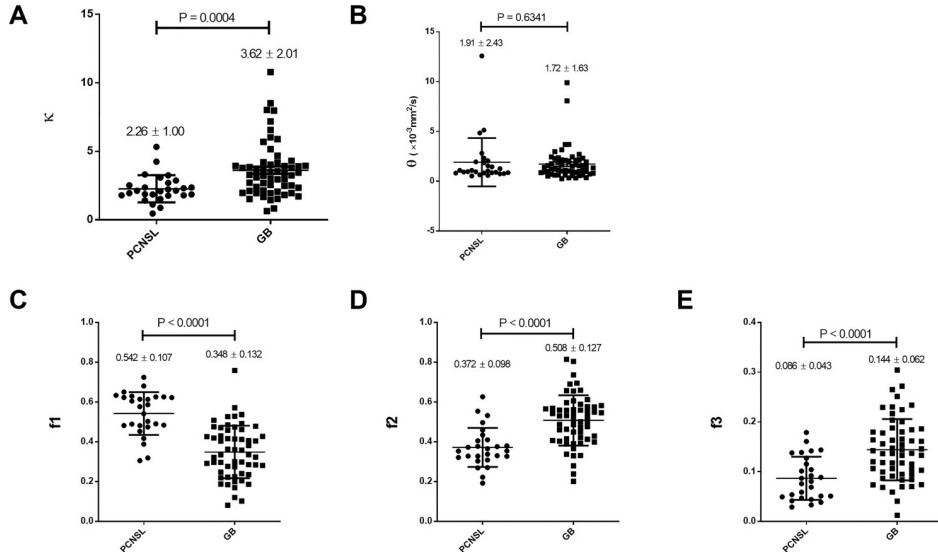

**Fig 2. Comparisons of the GD model-derived parameters between the PCNSLs and GBs in the gadolinium-enhancing lesion. A:** The κ was significantly smaller in the PCNSL group than in the GB group. **B:** The θ was not significantly different between the groups. **C–E:** The f1 was significantly larger and the f2 and f3 were significantly smaller in the PCNSL group than in the GB group.

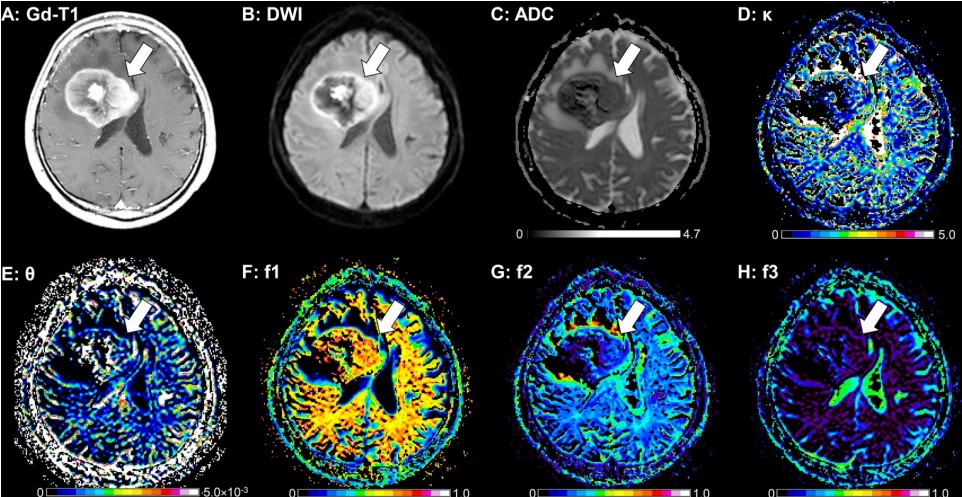

**Fig 3. A 62-year-old-male with a PCNSL. A:** The post-contrast T1-weighted image shows a ring-like enhancing mass lesion in the right frontal lobe (arrow). The enhancing lesion shows high signal intensity on the DWI (**B**) and a low ADC ($0.70 \times 10^{-3}$ mm$^2$/sec, **C**). This lesion shows a small $\kappa$ (1.76, **D**), a large $\theta$ ($4.85 \times 10^{-6}$ mm$^2$/sec, **E**), a large f1 (0.626, **F**), a small f2 (0.270, **G**), and a small f3 (0.104, **H**). The peritumoral T2-hyperintense area without contrast enhancement shows a large $\kappa$ (8.18, **D**), a small $\theta$ ($0.46 \times 10^{-6}$ mm$^2$/sec, **E**), a small f1 (0.139, **F**), a large f2 (0.772, **G**), and a small f3 (0.090, **H**).

The ADC values of the enhancing lesions were significantly smaller in the PCNSL group ($0.883 \pm 0.176 \times 10^{-3}$ mm$^2$/sec) compared to the GB group ($1.246 \pm 0.266 \times 10^{-3}$ mm$^2$/sec, p<0.0001). The PCNSL group's D values were significantly smaller ($0.805 \pm 0.167 \times 10^{-3}$ mm$^2$/sec) compared to the GB group's D values ($1.146 \pm 0.256 \times 10^{-3}$ mm$^2$/sec, p<0.0001). The D* was significantly smaller in the PCNSL group ($34.0 \pm 7.4 \times 10^{-3}$ mm$^2$/sec) versus the

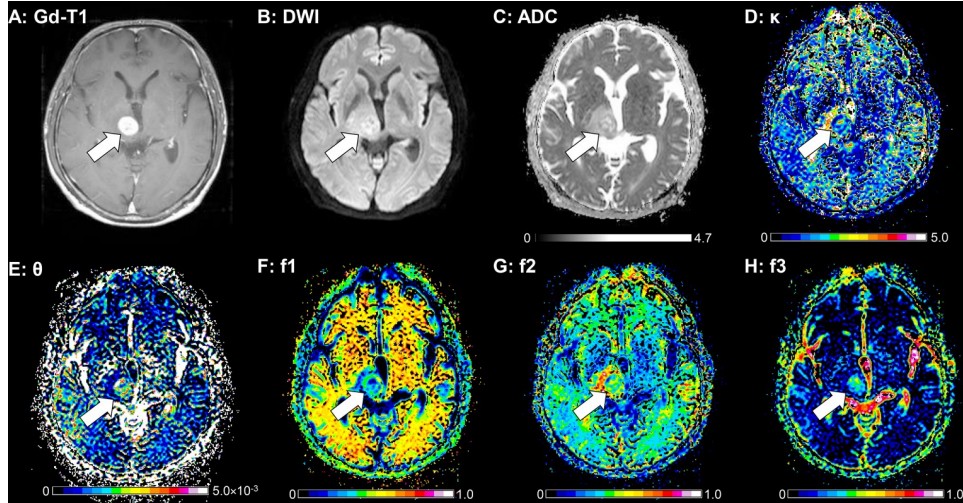

**Fig 4. A 66-year-old-male with a GB. A:** The post-contrast T1-weighted image shows a solid enhancing mass lesion in the right thalamus (arrow). The enhancing lesion shows partly high signal intensity on DWI (**B**) and a relatively high ADC ($1.42 \times 10^{-3}$ mm$^2$/sec, **C**). This lesion shows a small $\kappa$ (1.44, **D**), a large $\theta$ ($3.15 \times 10^{-6}$ mm$^2$/sec, **E**), a small f1 (0.297, **F**), a moderate f2 (0.399, **G**), and a large f3 (0.304, **H**). The peritumoral T2-hyperintense area without contrast enhancement shows a large $\kappa$ (3.94, **D**), a small $\theta$ ($0.75 \times 10^{-6}$ mm$^2$/sec, **E**), a small f1 (0.308, **F**), a large f2 (0.597, **G**), and a small f3 (0.095, **H**).

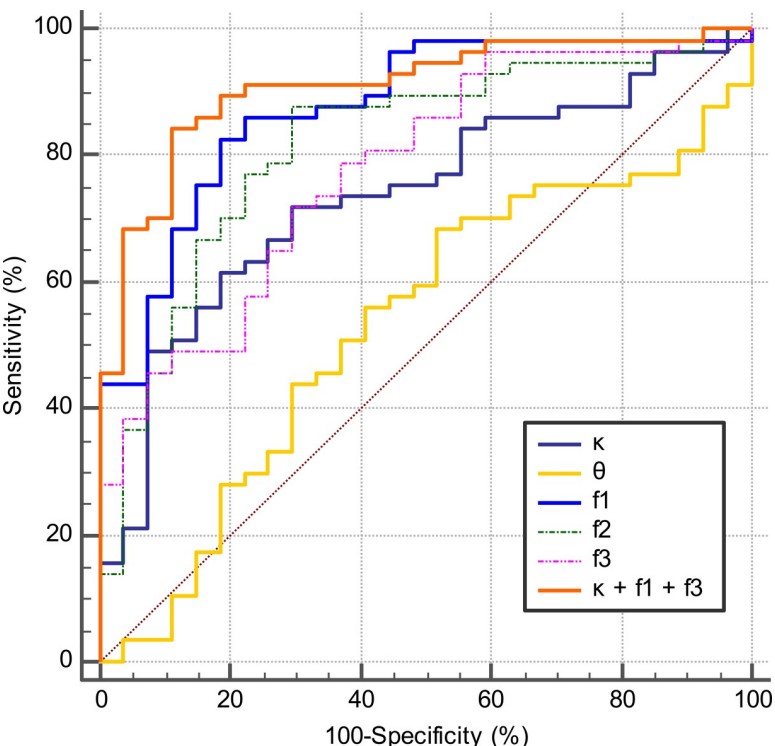

**Fig 5. ROC graphs.** The combination of κ, f1, and f3 demonstrated excellent diagnostic performance with the AUC of 0.909, sensitivity of 84.2%, and specificity of 88.9%. The f1 (AUC 0.877) and f2 (AUC 0.817) showed good performances. The κ (AUC 0.737) and f3 (AUC 0.778) showed fair diagnostic performances. The θ (AUC 0.533) resulted in a failed performance.

GB group ($40.7 \pm 5.6 \times 10^{-3}$ mm$^2$/sec, p<0.0001). The f was significantly smaller in the PCNSL group ($0.082 \pm 0.024$) compared to the GB group ($0.102 \pm 0.023$, p = 0.0005).

## Diagnostic performance of the single and combined parameters

The ROC graphs and diagnostic performance parameters are shown in **Fig 5** and **Table 2**. In the single-parameter analysis regarding the differential diagnosis of GBs and PCNSLs, the

**Table 2. ROC analysis for diagnostic performance of the parameters in the differentiation between PCNSLs from GBs.**

| Parameters | Area Under Curve | Sensitivity (%) | Specificity (%) | Cutoff Value | | |
|---|---|---|---|---|---|---|
| κ | 0.737 | 61.4 | 81.5 | 2.954 | | |
| θ | 0.533 | 68.4 | 48.1 | $0.971 \times 10^{-3}$ mm$^2$/sec | | |
| f1 | 0.877 | 82.5 | 81.5 | 0.474 | | |
| f2 | 0.817 | 87.7 | 70.4 | 0.380 | | |
| f3 | 0.778 | 71.9 | 70.4 | 0.104 | | |
| κ+f1+f3 | 0.909 | 84.2 | 88.9 | 0.540 | 0.404 | 0.133 |
| D | 0.875 | 86.0 | 81.5 | $0.887 \times 10^{-3}$ mm$^2$/sec | | |
| D* | 0.776 | 66.7 | 85.2 | $40.089 \times 10^{-3}$ mm$^2$/sec | | |
| f | 0.731 | 78.9 | 63.0 | 0.087 | | |
| D+f | 0.884 | 82.5 | 81.5 | $0.998 \times 10^{-3}$ | 0.072 | |
| ADC | 0.879 | 87.7 | 77.8 | $0.972 \times 10^{-3}$ mm$^2$/sec | | |

ROC, receiver operating characteristics; PCNSL, primary central nervous system lymphoma; GB, glioblastoma; D, true diffusion coefficient; D*, pseudo-diffusion coefficient; f, perfusion fraction; ADC, apparent diffusion coefficient.

ADC, f1, D, and f2 all showed good performances. The ADC showed the highest AUC value at 0.879, and the f1 and D values showed comparable AUCs (f1, 0.877; D, 0.875). No significant differences were found in the comparisons of ROC curves for these three parameters: f1 vs. ADC, p = 0.6130, f1 vs. D, p = 0.8449; ADC vs. D, p = 0.3935. The κ, f3, D*, and f showed fair diagnostic performances, but the θ resulted in a failed performance.

In the combined-parameters analysis, the stepwise procedure selected κ, f1, and f3 for the GD model, and the D and f for the IVIM model. The combination of κ, f1, and f3 revealed excellent diagnostic performance with the AUC of 0.909, sensitivity of 84.2%, and specificity of 88.9%. This combination increased the diagnostic performance of κ (p = 0.0016), and f3 (p = 0.0075), although it did not improve the performance of f1 (p = 0.1950). The AUC of this combination (0.909) was higher than that of ADC (0.879); however, there was no significant difference between them (p = 0.2152).

The combination of D and f showed good diagnostic performance with the AUC of 0.884, 82.5% sensitivity, and 81.5% specificity. This combination improved the diagnostic performance of f (p = 0.0077), although it did not improve the performance of D (p = 0.5276).

Among all of the single and combined parameters, the combination of κ, f1, and f3 showed the highest AUC; however, no significant differences were detected between this combination and the ADC (p = 0.2152) or the combination of D and f (p = 0.2207).

## Correlations of the model parameters

Fig 6 shows the correlations among the GD model-derived and IVIM model-derived parameters in all tumors. The f1 had an almost perfect inverse correlation with D (all, r = −0.9756, p<0.0001; PCNSL, r = −0.9558, p<0.0001; GB, r = −0.9699, p<0.0001). The f2 had a very strong positive correlation with D (all, r = 0.8865, p<0.0001; PNCSL, r = 0.9619, p<0.0001; GB, r = 0.8273, p<0.0001). The f3 had a very strong positive correlation with the f (all, r = 0.8654, p<0.0001; PNCSL, r = 0.8317, p<0.0001; GB, r = 0.8611, p<0.0001). The f1 had an very strong negative correlation with the f2 (all, r = −0.9155, p<0.0001; PCNSL, r = −0.9150, p<0.0001; GB, r = −0.8874, p<0.0001).

## Discussion

The results of our analyses revealed that in gadolinium-enhancing lesions, the κ was significantly smaller in the PCNSL group than in the GB group. The θ was not different between the groups. The f1 was larger, the f2 was smaller, and the f3 was lower in the PCNSLs than in the GBs. The low κ values observed in the PCNSLs indicated that the PDF curve had a right-skewed distribution, which meant that the PDF has its peak in the lower D area, and thus the fraction of lower D was larger. Since the θ values were not significantly different between the PCNSL and GB groups, it was likely that the lower κ values might result in the lower ADC and D values and the higher f1 values observed in the PCNSLs compared to the GBs. These findings are in accordance with studies that examined the mono-exponential model, in which PCNSLs showed lower ADC values relative to GBs [3–5].

The θ is a scale parameter and may thus reflect the heterogeneity of a biological tissue. We expected that the θ values would be larger in GBs than in PCNSLs since GBs are histologically characterized by intratumoral tissue heterogeneity whereas PCNSLs are characterized by the dense and homogenous distribution of tumor cells; however, no significant difference in the θ values was observed between the groups. The θ values showed large standard deviations in both the PCNSLs and the GBs, indicating that this value could vary widely even in the same type of tumor. The same trend was observed in a study of breast tumors in which the θ values

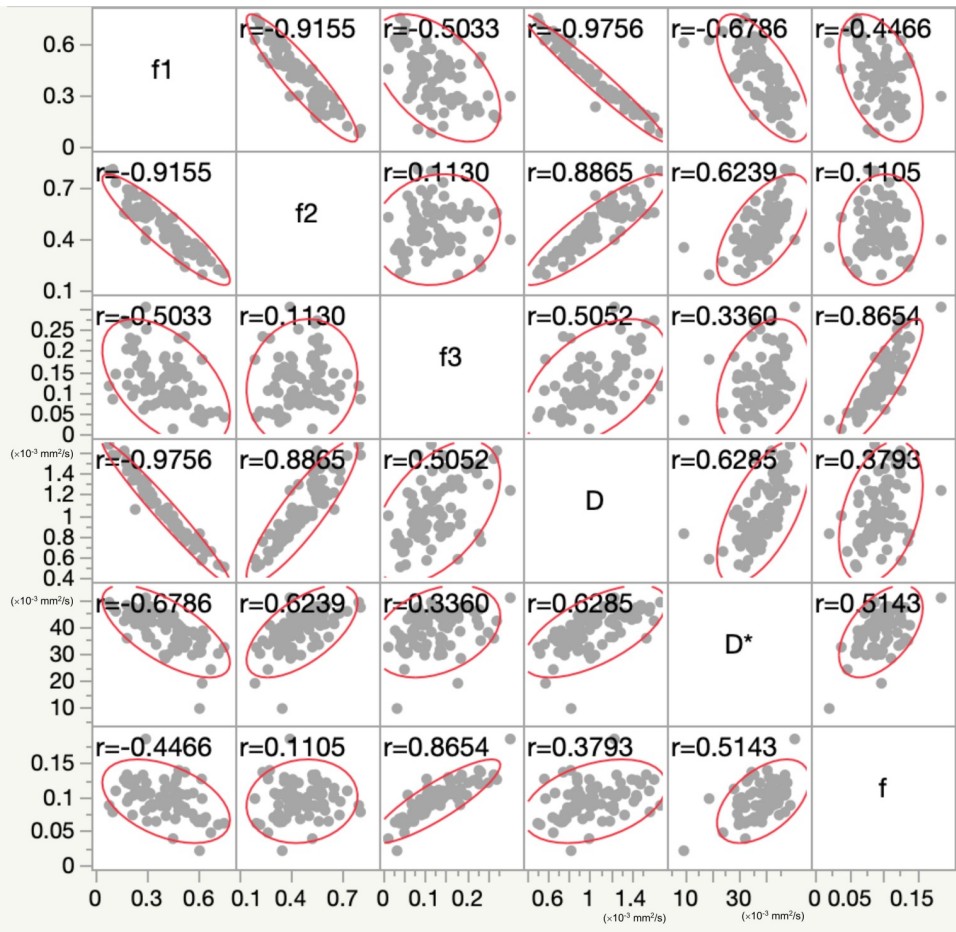

**Fig 6. The correlations among the GD model-derived and IVIM model-derived parameters in all tumors.** The f1 had an almost perfect inverse correlation with D. The f2 had a very strong positive correlation with D. The f3 had a very strong positive correlation with the f.

were not significantly different between the different types [18]. The utility of this parameter should be further evaluated in larger populations.

It seems that the higher f1 and lower f2 in the PCNSLs and the lower f1 and higher f2 in the GBs well reflected the histological features of the respective tumors. Histologically, PCNSLs are characterized by high cell density at the expense of reduced available extracellular space, and necrosis is not a common feature of this tumor. GBs can show locally high cell density, but the overall cell density can be lowered depending on the fraction of microscopic necrosis or hemorrhage. Our present findings are consistent with a study that reported that the ADC was lower and the cell density was higher in PNCSLs than in high-grade gliomas [3].

The GB group showed larger f3 and f compared to the PCNSL group. This may be attributed to the difference in vascularity of these tumors. Pathologically, neovascularization is a key feature of GB while it is not prominent in PCNSL [33, 34]. Our results are consistent with those from previous studies using dynamic susceptibility contrast perfusion-weighted imaging and arterial spin labeling imaging [9, 35].

With respect to the diagnostic performance, the ADC, f1, and D showed comparable AUCs in the present study, indicating that all three of these parameters are useful in the differentiation of PNCSLs and GBs. The reason for the slightly higher AUC observed with the ADC

could be the effect of perfusion on ADC measurements. In hyperperfused tissues, ADC will be affected by the perfusion effect and overestimated compared to D; however, since both f1 and D are parameters without a perfusion effect in theory, an overestimation caused by perfusion should not be observed in these values. Therefore, in hypervascular tumors such as GBs, the ADC should be larger than D. On the other hand, in hypovascular tumors such as PCNSLs, this difference between ADC and D should smaller. This means that the difference between ADC and D would be larger in GBs than in PCNSLs. Therefore, ADC could show higher diagnostic performance in the discrimination of these two tumors than D. In fact, the difference between the ADC and D values was greater in the GBs ($0.100 \times 10^{-3}$ mm$^2$/sec) than in the PCNSLs ($0.078 \times 10^{-3}$ mm$^2$/sec), which was most likely due to the higher perfusion effect on the ADC in GBs than in PCNSLs. Nevertheless, the combination of κ, f1, and f3 demonstrated the highest diagnostic performance among all of the single and combined parameters, with the AUC of 0.909. The AUC of this combination tended to be higher than that of ADC although there was no statistically significant difference. Whether the combination of parameters of the GD model has an additive value should be evaluated in a larger population, since we did not observe statistical significance in all of our comparisons.

We found the correlations between the GD model-derived and IVIM-derived parameters, particularly between the f1 and D, the f2 and D, and the f3 and f. The almost perfect negative correlation observed between the f1 and D may indicate that these two parameters contain virtually identical information. The positive correlation between f2 and D suggests that the increased extracellular space like that taken up by microscopic necrosis might result in the higher f2. The positive correlation between f3 and f indicates that both of these parameters well reflected tissue perfusion despite the different analysis methods used. The negative correlation between f1 and f2 was likely due to the complementary relationship between these two parameters. In general, intravascular space ($\doteqdot$ f3) is smaller compared to intracellular ($\doteqdot$ f1) and extracellular extravascular space ($\doteqdot$ f2). In fact, the f3-values were much smaller than the f1- and f2-values in both PCNSLs and GBs in the present study. Therefore, the increase in f1 would result in the decrease in f2, and vice versa. Although the GD-derived and IVIM-derived parameters provide similar information, the strength of the GD model-derived parameters is that all fraction values (f1, f2, f3) are expressed as fractions or percentages, which allows us to well characterize tumors from histological viewpoint. The IVIM-derived f-value is also expressed in a percentage or fraction; however, the IVIM analysis is not able to provide the fraction values for intracellular and extracellular-extravascular spaces. In this sense, the IVIM method is not a perfect method for the histological characterization of tumors.

In the T2-hyperintense lesions without contrast enhancement, no significant differences were observed between the PCNSL and GB groups for any parameters. There have been several studies that showed increased rCBV on DSC-perfusion imaging in peritumoral noncontrast-enhancing T2-hyperintense areas of GBs [36, 37]. The results of these studies indicated that the peritumoral areas of GB include not only vasogenic edema but also tumor cells infiltrating surrounding brain parenchyma; however, our study did not reveal any significant differences in the GD model-based parameters for peritumoral noncontrast-enhancing T2-hyperintense areas between PCNSLs and GBs. The f2 values in the noncontrast-enhancing T2-hyperintense areas were higher in both types of tumor compared to those in the contrast-enhancing areas and normal appearing white matter. We assume that the high f2 values in the noncontrast-enhancing T2-hyperintense areas are likely to reflect mostly perifocal vasogenic edema rather than tumor infiltration outside the enhancing lesion. Our result is consistent with the previous DWI study in which ADC could not be used to differentiate edema with infiltration of tumor cells from vasogenic edema in high-grade gliomas and PCNSLs [38].

In the normal-appearing white matter, the GB group showed larger f1, smaller f2, and larger f3 than the PCNSL group although these differences were small. This was unexpected, and the reasons for the differences remain unclear; however, since GBs frequently show extensive infiltration into the surrounding brain tissue, which is a fundamental feature of diffuse glioma, it is no wonder that the increased cell density and perfusion were observed in the normal-appearing white matter.

This study has several limitations. The number of patients was relatively small (n = 84) — especially the number of patients with PCNSL (n = 27). The only one person performed the ROI placements on a single slice, and not whole tumor volume was evaluated. The ROI placements on the gadolinium-enhancing lesions were occasionally difficult, particularly when the lesions showed irregular and thin ring-like enhancement. Although the best effort was made to include only enhancing lesions, it is possible that necrosis in tumors was included, and this could have affected the analyses. In addition, the selection of b-values has not yet been optimized. Prior studies of the GD model used the maximum b-values ranging from 1000 to 3000 sec/mm$^2$ [18, 22–24]. In a study of prostate cancers, Oshio et al. used the similar DWI parameters to ours and the highest b-value of 1000 s/mm$^2$, and reported that the good fitting accuracy was observed in both cancerous tissues ($R^2$ = 0.99226) and normal tissues ($R^2$ = 0.99842) [22]. Their result indicated that DWI with the highest b-value of 1000 s/mm$^2$ can be used for GD model analyses; however, since it was reported that the non-monoexponential diffusion-related signal decay generally becomes more obvious over more extended b-value ranges, the maximum b value of 1000 sec/mm$^2$ used in the present study might be lower than the optimal value. The optimal b-values and numbers should be elucidated in future studies.

## Conclusions

The GD model well described the histological features of PCNSLs and GBs, and its use enabled the significant differentiation of these tumors. The κ, f2, and f3 values were significantly smaller and the f1 values were significantly larger in the PCNSLs than in the GBs. The combination of κ, f1, and f3 showed the highest AUC. The GD model-derived parameters correlated well with the IVIM-derived parameters. The GD model may therefore contribute to the characterization of various brain tumors from the histological viewpoint.

## Supporting information

**S1 Data. All measurements for gamma distribution model-derived and IVIM model-derived parameters.**
(XLSX)

## Author Contributions

**Conceptualization:** Osamu Togao, Toru Chikui, Akio Hiwatashi.

**Data curation:** Osamu Togao, Yukiko Kami, Kazufumi Kikuchi, Daichi Momosaka, Yoshitomo Kikuchi, Daisuke Kuga, Nobuhiro Hata, Masahiro Mizoguchi, Koji Iihara, Akio Hiwatashi.

**Formal analysis:** Osamu Togao.

**Funding acquisition:** Osamu Togao.

**Investigation:** Osamu Togao.

**Methodology:** Osamu Togao, Toru Chikui, Kenji Tokumori, Yukiko Kami.

**Project administration:** Osamu Togao.

**Resources:** Osamu Togao.

**Software:** Osamu Togao, Toru Chikui, Yukiko Kami.

**Supervision:** Osamu Togao, Toru Chikui, Koji Iihara, Akio Hiwatashi.

**Validation:** Osamu Togao.

**Visualization:** Osamu Togao.

**Writing – original draft:** Osamu Togao, Toru Chikui, Kenji Tokumori, Yukiko Kami, Kazufumi Kikuchi, Daichi Momosaka, Yoshitomo Kikuchi, Daisuke Kuga, Nobuhiro Hata, Masahiro Mizoguchi, Koji Iihara, Akio Hiwatashi.

**Writing – review & editing:** Osamu Togao, Toru Chikui, Kenji Tokumori, Yukiko Kami, Kazufumi Kikuchi, Daichi Momosaka, Yoshitomo Kikuchi, Daisuke Kuga, Nobuhiro Hata, Masahiro Mizoguchi, Koji Iihara, Akio Hiwatashi.

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
