## [Decision Letter · Decision Letter 0]

10 Jul 2020

PONE-D-20-15410

Gamma distribution model of diffusion MRI for the differentiation of primary central
nerve system lymphomas and glioblastomas

PLOS ONE

Dear Dr. Togao,

Thank you for submitting your manuscript to PLOS ONE. After careful consideration, we
feel that it has merit but does not fully meet PLOS ONE’s publication criteria as it
currently stands. Therefore, we invite you to submit a revised version of the
manuscript that addresses the points raised during the review process.

Please do your best to address all of the comments, particularly pay attention to
those where references have been missed, as per Reviewer 1's comments. I strongly
recommend that you follow each of the suggestions provided by the Reviewers in order
to increase the overall readability and interpretation of the results.

Please submit your revised manuscript by Aug 24 2020 11:59PM. If you will need more
time than this to complete your revisions, please reply to this message or contact
the journal office at plosone@plos.org. When
you're ready to submit your revision, log on to https://www.editorialmanager.com/pone/ and select the 'Submissions
Needing Revision' folder to locate your manuscript file.

If you would like to make changes to your financial disclosure, please include your
updated statement in your cover letter. Guidelines for resubmitting your figure
files are available below the reviewer comments at the end of this letter.

We look forward to receiving your revised manuscript.

Kind regards,

Niels Bergsland

Academic Editor

PLOS ONE

Journal Requirements:

2. Thank you for including your ethics statement: 'This retrospective study was
approved by the Institutional Review Board of our hospital (no. 2019-447), and the
requirement for informed consent was waived.'   

(a) Please amend your current ethics statement to include the full name of the ethics
committee/institutional review board(s) that approved your specific study.  

(b) Once you have amended this/these statement(s) in the Methods section of the
manuscript, please add the same text to the “Ethics Statement” field of the
submission form (via “Edit Submission”).

3. In the ethics statement in the manuscript and in the online submission form,
please provide additional information about the patient records used in your
retrospective study, including: a) whether all data were fully anonymized before you
accessed them; b) the date range (month and year) during which patients' medical
records were accessed; and c) the source of the medical records analyzed in this
work (e.g. hospital, institution or medical center name).

Reviewers' comments:

Reviewer's Responses to Questions

**Comments to the Author**

1. Is the manuscript technically sound, and do the data support the conclusions?

Reviewer #1: Yes

Reviewer #2: Yes

2. Has the statistical analysis been performed
appropriately and rigorously? 

Reviewer #1: Yes

Reviewer #2: Yes

3. Have the authors made all data underlying the
findings in their manuscript fully available?

Reviewer #1: No

Reviewer #2: Yes

4. Is the manuscript presented in an intelligible
fashion and written in standard English?

Reviewer #1: Yes

Reviewer #2: Yes

5. Review Comments to the Author

Reviewer #1: ----Summary

The manuscript reports the use of the diffusion MRI gamma distribution (GD) model in
the differentiation of primary central nervous system lymphomas (PCNSLs) and
glioblastomas (GBs).

The results indicate that the GD model is advantageous in terms of diagnostic
performance when comparing these 2 lesions, namely when 3 specific parameters of
this model are used in combination.

My overall impression is that the manuscript has quality and it is an interesting an
innovative topic, but some aspects should be considered and better analysed.

----Comments to the Author

• Abstract:

Page 2 - “In this study, we investigate whether whether the gamma distribution (GD)
model is useful in this differentiation of PNCSLs. and GBs” – The word “whether” is
repeated and the full stop after PNCSLs should be removed.

Page 2 - “Receiver operating curve (ROC) analyses were performed to assess diagnostic
performance.” – ROC stands for Receiver Operating Characteristic. This should appear
as: Receiver Operating Characteristic (ROC) curve. Please check the rest of the
document, as this is recurrent.

• Introduction:

Page 4 – Last paragraph – “The bi-exponential intravoxel incoherent motion (IVIM)
model aims to separate the true water diffusion and the capillary perfusion by using
multiple b-values [10, 11].” – multiple b-values but specifically multiple low
b-values. Please add the word “low”

There are only a few articles in this topic, and the way the authors composed parts
of the introduction is very similar to the reference number 22 (Shinmoto et al.
2014) and/or 24 (Borlinhas et al. 2019 ) of the document, and sometimes being
exactly the same, but not being cited.

Example 1

“The IVIM model has an associated uncertainty to the estimated pseudodiffusion, and
perfusion fraction, and a possible overparametrization of the model.7 The
limitations of the DKI model are related to the unclear biological interpretation of
the mean kurtosis parameter, and to the effects of the high b-values that the model
requires.” – Reference 24 of the manuscript

“The bi-exponential model could be influenced by an uncertainty of the estimated
perfusion, since signal measurements at low b-values are susceptible to measurement
errors [15-17]. The DKI model is limited by the unclear biological interpretation of
the kurtosis parameters [18-20].” – The actual text in the manuscript

Example 2

“The GD model allows us to estimate fractions of a tissue type based on the concept
that the area fractions for D <1.0 × 10−3 mm2/sec, 1.0 × 10−3 to 3.0 ×10−3
mm2/sec, and D >3.0 ×10−3 mm2/sec are attributed to intracellular, extracellular
extravascular, and intravascular spaces, respectively [21].” – The actual text in
the manuscript

– “ADC values smaller than 1.0 mm2/scan be attributed to small tumor cells with
restricted diffusion, and ADC values larger than 3.0 mm2/s can be attributed to
perfusion, with ADC values between 1.0 mm2/sand 3.0 mm2/s attributed to water
diffusion in the other components.” – Reference 21 of the manuscript

– “(…)extracellular fluid (ADCs between 1.0 and 3.0 mm2/s)" – Reference 22 of the
manuscript

– “Through the PDF of ADC, three different areas under the function’s curve are
defined as follows: the fraction of diffusion lower than 1.00 × 10-3 mm2/s is the f1
fraction and it reflects the small cell component; the fraction of diffusion higher
than 3.00 × 10-3 mm2/s is the f3 fraction and it reflects the perfusion component;
and the fraction of diffusion between 1.00 × 10-3 mm2/s and 3.00 × 10-3 mm2/s is the
f2 fraction translating the extracellular component of the tissue.12,13” - Reference
24 of the manuscript

In this case, and as you can see, reference 21 is not specific when mentioning the
meaning of f2. You have more specific references such as number 22 and 24 of your
document.

Please add these references.

Page 5 - “The GD model allows us to estimate fractions of a tissue type based on the
concept that the area fractions for D <1.0 × 10−3 mm2/sec, 1.0 × 10−3 to 3.0
×10−3 mm2/sec, and D >3.0 ×10−3 mm2/sec are attributed to intracellular,
extracellular extravascular, and intravascular spaces, respectively [21]” – This
part “1.0 × 10−3 to 3.0 ×10−3 mm2/sec” must appear as “D=1.0 × 10−3 to 3.0 ×10−3
mm2/sec”

Page 5 – “The GD model is suitable for realistically interpreting diffusion data in a
histological context.” - This sentence is very abrupt. If the reference you are
using is the reference 21, Oshio et al. cautiously concluded the following: “(…)
histological interpretation of the data appears possible.” In Oshio et al. work, the
peripheral zone of the prostate and a prostate cancer were being compared, not
different histological types of lesions for example. Your sentence can be
misinterpreted.

Also, this sentence that you present should be connect to the information included in
this paragraph.

Page 5 – “The GD model has been used to assess prostate cancers [21-23], breast
cancers [24], and renal function [25]. To the best of our knowledge, its application
to brain tumours has never been reported. We conducted the present study to
determine whether the GD model is useful in the differentiation of PCNSLs and GBs.”
– To the best of my knowledge that is true, but the GD model has been applied to
brain studies. This is a relatively new application, and consequently, it would be
worth mentioning this fact. Here is the a reference that you can use: Grinberg F,
Farrher E, Ciobanu L, Geffroy F, Le Bihan D, et al. (2014) Non-Gaussian Diffusion
Imaging for Enhanced Contrast of Brain Tissue Affected by Ischemic Stroke. PLoS ONE
9(2): e89225. doi:10.1371/journal.pone.0089225

• Material and Methods

- Patients

Page 6 - “Multi-b-value DWI” – It would be more adequate to read for example “The use
of a DWI protocol with multiple b-values (…)”

Page 6 – The difference between the number of PCNSLs and GBs is relevant. Knowing
that the PCNSLs are less frequent when compared to the GBs, this information could
be added to justify the difference in the groups.

If available, the characteristics of the GB, like fraction of necrosis or haemorrhage
should be reported. The addition of these characteristics would be enriching the
study and could explain some outliers in the results.

- ROI placement

Page 9 – The legend of this figure 1 repeats the information that it is presented in
the text: “The ROIs were also placed on the noncontrast-enhancing T2-hyperintense
areas as well as on the contralateral normal-appearing white matter on the image
obtained with the b-value of 0 sec/mm2 image.”

The reason why and the purpose of placing ROIs on the noncontrast-enhancing
T2-hyperintense areas should be stated.

- Statistical Analysis

Page 9 – “The optimal cutoff point was determined by Youden's method.” – A literature
reference should be provided to support the method.

Page 10 – “To determine whether the combination of multiple parameters for both the
GD model and the IVIM model increases the diagnostic performance, we first performed
a stepwise analysis to select appropriate parameters, and then we performed a
binomial logistic regression analysis to examine the combinations of the selected
parameters.” – Does “appropriate” means with the best diagnostic performances?
“Examine” in what way? This should be presented in a clearer way.

• Results

- Comparisons of the parameters between the PCNSL and GB groups

Page 10 - 11 - “Figure 2 illustrates the results of our comparisons of the GD
model-derived parameters between the PCNSLs and GBs in the gadolinium-enhancing
lesions. The κ was significantly smaller in the PCNSL group (2.26 ± 1.00) than in
the GB group (3.62 ± 2.01, p=0.0004). The θ was not significantly different between
the groups. The f1 was significantly larger in the PCNSL group (0.542 ± 0.107) than
in the GB group (0.348 ± 0.132, p<0.0001). The f2 was significantly smaller in
the PCNSL group (0.372 ± 0.098) than in the GB group (0.508 ± 0.127, p<0.0001).
The f3 was also significantly smaller in the PCNSL group (0.086 ± 0.043) than in the
GB group (0.144 ± 0.062, p<0.0001). The detailed information for the parameters
is summarized in Table 1. Figure 3 provides a PCNSL case that showed a ring-like
enhancing mass lesion mimicking a GB. This lesion showed a low κ, a large f1, a
small f2, and a small f3, suggesting PCNSL. Figure 4 demonstrates a GB case that
showed a solid enhancing mass lesion. This lesion showed a small κ, a small f1,
moderate f2 and large f3, which are consistent with GB. In the T2-hyperintense areas
without contrast enhancement, no significant differences were found between the
PCNSL and GB groups for any of the GD model derived parameters.” - This paragraph
should be written in a more fluid way so the reader can better understand the ideas
that the authors chose to highlight.

Page 14 – The results T2-hyperintense areas without contrast enhancement were
mentioned but no analysis was made to the NAWM results. If the results were obtained
these should be used to support the information shown about the GD model in the
manuscript.

Page 14 – In the "all data" excel sheet for f parameter you use percentage and here
you show a fraction. Please make it uniform to clearly inform the reader.

Page 18 – “(…) although the difference was not significantly different from the
values for D (p=0.5276).” – This should be rephrased.

- Correlations of the model parameters

Page 18 – “The f1 had an almost perfect inverse correlation with D (all,

r = −0.976, p<0.0001; (…)” – According to figure 6, r = −0.9756, please confirm
the information and change where necessary.

Page 18 – Considering that the correlation between f2 and D parameters is 0.88657,
and it is mentioned because it is high compared to the other correlations, the
correlation between f1 and f2 should also be mentioned. The correlation between f1
and f2 is above 0.9155 and the meaning of this result can be of interest to be
discussed in the appropriate section of this work.

• Discussion

- The meaning of f3 and f parameters results and finding should be analysed in the
discussion. Why f and f3 parameters are higher in the PCNSL when compared to the GB
group of lesions?

- Another important result, which should be discussed, is that with the GD model
parameters (κ, f1, and f3) it was possible to obtain a higher AUC when compared to
ADC’s AUC. Can the author provide a justification or further analysis?

Page 20 – “The reason for the slightly higher AUC observed with the ADC could be the
effect of perfusion on ADC measurements.” - Can the effect of perfusion on ADC
explain the higher AUC observed? If that is the case, in what way can this be
explained?

Page 21 – “We found the correlations between the GD model-derived and IVIM-derived
parameters.” – This sentence is vague or incomplete please consider revising it.

Page 21 – “The almost perfect correlation observed between the f1 and D may indicate
that these two parameters contain virtually identical information.” – It would be
important to mention that it is a negative correlation.

Page 21 – “The positive correlation between f2 and D suggests the opposite, and the
increased extracellular space like that taken up by microscopic necrosis might
result in the higher f2.” – The “opposite” to what?

Page 21 – “Although the GD-derived and IVIM-derived parameters provide similar
information, the strength of the GD model-derived parameters is that the values are
expressed as a fraction or percentage, which allows us to characterize

tumors from histological viewpoint.” - “f “ is an IVIM-derived parameter and it is
also expressed in percentage or fraction.

Page 21 – “The high f2 values in both types of tumor are likely to reflect mostly
perifocal vasogenic edema rather than tumor infiltration outside the enhancing
lesion” – “high f2 values” relative to what? The sentence would be more specific if
it mentioned that it is referring to T2-hyperintense lesions. This idea would
benefit from an addition of a literature reference.

Page 22 – Another limitation is the fact that only one person performed the ROI
placement. The fact that the highest b-value in use was 1000s/mm2, should be better
justified since in the brain higher b-values are usually used.

- Figures

Figure 1. It would be interesting to have the non-contrast-enhancing T2 image where
the ROIs were also placed.

Figure 5 The y axis title is missing

Figure 6 Units should be included, when appropriate.

- Tables

Table 1 It would be easier to interpret the information shown in the table if
p-values were also presented.

Table 2 In this table the reader is first exposed to the combination of parameters
that were used to estimate diagnostic performances, additionally to the estimation
of the diagnostic performances with the individual parameters. Why these specific
combinations of parameters were considered? The justification should be clearly
stated in the text.

In the legend “D, true diffusion coefficient; D*, ç; f, perfusion fraction” the
meaning of D* is missing.

- Supporting information

Units should be added to the “all data” table where appropriate.

----Statement:

The topic is interesting but there are some parts of the study which the description
should be improved so the reader can better understand it.

The major strengths of the article are: the application of a new diffusion model to
brain tumours, which as far as I know it has not been done yet; the inclusion of an
analysis of combined parameters and the evaluation of its performance; the inclusion
of healthy tissue results, but these results were not compared to the results obtain
for tumours which is a weakness. Consequently, the relevant weaknesses of the work
are: the need for a deeper analysis of the obtained results; there are some methods
and procedures that should be better described and justified; the use of b=1000
s/mm2 as the highest b-value is unusual in a brain diffusion studies, and this
should be further justified.

Note: “Have the authors made all data underlying the findings in their manuscript
fully available?” My answer was “no” because I only had access to the summary
statistics, and not to the data points behind the statistics.

Reviewer #2: This study examined gamma distribution model of diffusion MRI and this
model is useful to differentiate malignant lymphoma and glioblastoma.

This is well written paper and there are some minor points to revise.

In table 1, clarify the statistical significant differences between PCNSL and GB.

In figure 4, the k and f2 values at lateral peritumor area (lateral side) are high
compared to contrast-enhanced tumor area.

Discussion

Limitation. The authors evaluate the only one slice of tumor, not whole tumor volume.
Add this point in the limitation.

6. PLOS authors have the option to publish the peer
review history of their article (what does this mean?). If published, this will
include your full peer review and any attached files.

If you choose “no”, your identity will remain anonymous but your review may still be
made public.

**Do you want your identity to be public for this peer review?** For
information about this choice, including consent withdrawal, please see our
Privacy Policy.

Reviewer #1: No

Reviewer #2: No

---

## [Author Response · Author response to Decision Letter 0]

7 Sep 2020

First of all, we would like to thank the two reviewers for their thorough reading and
constructive criticism which greatly enhanced the quality of the paper. According to
the comments, we have carefully revised the paper.

Reviewer #1:----Summary

The manuscript reports the use of the diffusion MRI gamma distribution (GD) model in
the differentiation of primary central nervous system lymphomas (PCNSLs) and
glioblastomas (GBs).

The results indicate that the GD model is advantageous in terms of diagnostic
performance when comparing these 2 lesions, namely when 3 specific parameters of
this model are used in combination.

My overall impression is that the manuscript has quality and it is an interesting an
innovative topic, but some aspects should be considered and better analysed.

----Comments to the Author

• Abstract:

Page 2 - “In this study, we investigate whether whether the gamma distribution (GD)
model is useful in this differentiation of PNCSLs. and GBs” – The word “whether” is
repeated and the full stop after PNCSLs should be removed.

Response: Thank you for pointing out. These have been eliminated.

Page 2 - “Receiver operating curve (ROC) analyses were performed to assess diagnostic
performance.” – ROC stands for Receiver Operating Characteristic. This should appear
as: Receiver Operating Characteristic (ROC) curve. Please check the rest of the
document, as this is recurrent.

Response: Thank you very much. This has been corrected.

• Introduction:

Page 4 – Last paragraph – “The bi-exponential intravoxel incoherent motion (IVIM)
model aims to separate the true water diffusion and the capillary perfusion by using
multiple b-values [10, 11].” – multiple b-values but specifically multiple low
b-values. Please add the word “low”

Response: We agree with this point. The term “low” has been added here.

There are only a few articles in this topic, and the way the authors composed parts
of the introduction is very similar to the reference number 22 (Shinmoto et al.
2014) and/or 24 (Borlinhas et al. 2019 ) of the document, and sometimes being
exactly the same, but not being cited.

Example 1

“The IVIM model has an associated uncertainty to the estimated pseudodiffusion, and
perfusion fraction, and a possible overparametrization of the model.7 The
limitations of the DKI model are related to the unclear biological interpretation of
the mean kurtosis parameter, and to the effects of the high b-values that the model
requires.” – Reference 24 of the manuscript

“The bi-exponential model could be influenced by an uncertainty of the estimated
perfusion, since signal measurements at low b-values are susceptible to measurement
errors [15-17]. The DKI model is limited by the unclear biological interpretation of
the kurtosis parameters [18-20].” – The actual text in the manuscript

Response: The reference #18 (previous #24, Borlinhas et al.) has been added and cited
for these sentences as suggested.

Example 2

“The GD model allows us to estimate fractions of a tissue type based on the concept
that the area fractions for D <1.0 × 10−3 mm2/sec, 1.0 × 10−3 to 3.0 ×10−3
mm2/sec, and D >3.0 ×10−3 mm2/sec are attributed to intracellular, extracellular
extravascular, and intravascular spaces, respectively [21].” – The actual text in
the manuscript

– “ADC values smaller than 1.0 mm2/scan be attributed to small tumor cells with
restricted diffusion, and ADC values larger than 3.0 mm2/s can be attributed to
perfusion, with ADC values between 1.0 mm2/sand 3.0 mm2/s attributed to water
diffusion in the other components.” – Reference 21 of the manuscript

– “(…)extracellular fluid (ADCs between 1.0 and 3.0 mm2/s)" – Reference 22 of the
manuscript

– “Through the PDF of ADC, three different areas under the function’s curve are
defined as follows: the fraction of diffusion lower than 1.00 × 10-3 mm2/s is the f1
fraction and it reflects the small cell component; the fraction of diffusion higher
than 3.00 × 10-3 mm2/s is the f3 fraction and it reflects the perfusion component;
and the fraction of diffusion between 1.00 × 10-3 mm2/s and 3.00 × 10-3 mm2/s is the
f2 fraction translating the extracellular component of the tissue.12,13” - Reference
24 of the manuscript.

In this case, and as you can see, reference 21 is not specific when mentioning the
meaning of f2. You have more specific references such as number 22 and 24 of your
document.

Please add these references.

Response: We have added the two references (Borlinhas F, et al. and Shinmoto H, et
al.) for the sentence “The GD model allows us to estimate fractions of a tissue type
based on the concept that the area fractions for D <1.0 × 10−3 mm2/sec,
1.0 × 10−3 to 3.0 ×10−3 mm2/sec, and D >3.0 ×10−3 mm2/sec are attributed to
intracellular, extracellular extravascular, and intravascular spaces, respectively
[18, 22, 23]” 

Page 5 - “The GD model allows us to estimate fractions of a tissue type based on the
concept that the area fractions for D <1.0 × 10−3 mm2/sec, 1.0 × 10−3 to 3.0
×10−3 mm2/sec, and D >3.0 ×10−3 mm2/sec are attributed to intracellular,
extracellular extravascular, and intravascular spaces, respectively [21]” – This
part “1.0 × 10−3 to 3.0 ×10−3 mm2/sec” must appear as “D=1.0 × 10−3 to 3.0 ×10−3
mm2/sec”

Response: This has been corrected to “D =1.0 × 10−3 to 3.0 ×10−3 mm2/sec” as
suggested.

Page 5 – “The GD model is suitable for realistically interpreting diffusion data in a
histological context.” - This sentence is very abrupt. If the reference you are
using is the reference 21, Oshio et al. cautiously concluded the following: “(…)
histological interpretation of the data appears possible.” In Oshio et al. work, the
peripheral zone of the prostate and a prostate cancer were being compared, not
different histological types of lesions for example. Your sentence can be
misinterpreted.

Also, this sentence that you present should be connect to the information included in
this paragraph.

Response: We agree with this point. We want to state that these fractions (f1, f2,
f3) allow us to estimate histological conditions of tumors and organs. For example,
tumors with high f1, f2, and f3 should have high cell density, large interstitial
space, and high vascularity, respectively. So, this sentence has been replaced by
the following one. We think this sentence is now natural in the context.

Introduction (Page 5, line 17-18)

“Based on these fractions, we may be able to estimate histopathological conditions of
neoplasms or organs.”

Page 5 – “The GD model has been used to assess prostate cancers [21-23], breast
cancers [24], and renal function [25]. To the best of our knowledge, its application
to brain tumours has never been reported. We conducted the present study to
determine whether the GD model is useful in the differentiation of PCNSLs and GBs.”
– To the best of my knowledge that is true, but the GD model has been applied to
brain studies. This is a relatively new application, and consequently, it would be
worth mentioning this fact. Here is the a reference that you can use: Grinberg F,
Farrher E, Ciobanu L, Geffroy F, Le Bihan D, et al. (2014) Non-Gaussian Diffusion
Imaging for Enhanced Contrast of Brain Tissue Affected by Ischemic Stroke. PLoS ONE
9(2): e89225. doi:10.1371/journal.pone.0089225

Response: Thank you very much for the reference. We have modified the paragraph as
follows. We would like to keep the sentence “To the best of our knowledge...” since
the application to brain “tumors” have never been reported yet.

Introduction (Page 5, Line 20 – Page 6, Line 2)

“The GD model has been used to assess prostate cancers [22-24], breast cancers [18],
and renal function [25]. The GD model was also used to assess cerebral ischemic
stroke in rat brains, and it was showed that this model exhibited a better
performance than the conventional mono-exponential model and allowed for a
significantly enhanced visualization of ischemic lesions [26]. To the best of our
knowledge, its application to brain tumors has never been reported. We conducted the
present study to determine whether the GD model is useful in the differentiation of
PCNSLs and GBs.”

New reference

26. Grinberg F, Farrher E, Ciobanu L, Geffroy F, Le Bihan D, Shah NJ. Non-Gaussian
diffusion imaging for enhanced contrast of brain tissue affected by ischemic stroke.
PLoS One. 2014;9(2):e89225. Epub 2014/03/04. doi: 10.1371/journal.pone.0089225.
PubMed PMID: 24586610; PubMed Central PMCID: PMCPMC3937347.

• Material and Methods

- Patients

Page 6 - “Multi-b-value DWI” – It would be more adequate to read for example “The use
of a DWI protocol with multiple b-values (…)”

Response: Thank you for your suggestion. We have changed the wording as follows.

Materials and Methods (Page 6, Line 9, 11)

“The DWI protocol with multiple b-values has been a part of our routine preoperative
MRI examination for patients with brain tumors since January 2013.”

“The DWI with multiple b-values was conducted preoperatively for the patient during
the period from January 2013 to August 2019.”

Page 6 – The difference between the number of PCNSLs and GBs is relevant. Knowing
that the PCNSLs are less frequent when compared to the GBs, this information could
be added to justify the difference in the groups.

Response: Yes, the MLs are less frequent compared to GBs. The following sentence and
a new reference have been added.

Materials and Methods (Page 6, Line 20-21)

“The difference between the number of patients with PCNSLs and GBs can be explained
by the fact that the PCNSLs are less frequent compared to the GBs [27].” 

New reference

27. Ostrom QT, Cioffi G, Gittleman H, Patil N, Waite K, Kruchko C, et al. CBTRUS
Statistical Report: Primary Brain and Other Central Nervous System Tumors Diagnosed
in the United States in 2012-2016. Neuro Oncol. 2019;21(Suppl 5):v1-v100. Epub
2019/11/02. doi: 10.1093/neuonc/noz150. PubMed PMID: 31675094; PubMed Central PMCID:
PMCPMC6823730.

If available, the characteristics of the GB, like fraction of necrosis or haemorrhage
should be reported. The addition of these characteristics would be enriching the
study and could explain some outliers in the results.

Response: Thank you very much for your kind suggestion. But, in the present study, we
excluded necrosis or hemorrhage in the ROI-based measurements since we would like to
evaluate tumorous lesions. It is possible to report the fraction of necrosis or
frequency of hemorrhage, but I am afraid it may be confusing for readers since we
avoided the areas with necrosis or hemorrhage in the measurements.

- ROI placement

Page 9 – The legend of this figure 1 repeats the information that it is presented in
the text: “The ROIs were also placed on the noncontrast-enhancing T2-hyperintense
areas as well as on the contralateral normal-appearing white matter on the image
obtained with the b-value of 0 sec/mm2 image.”

The reason why and the purpose of placing ROIs on the noncontrast-enhancing
T2-hyperintense areas should be stated.

Response: The legend of Figure 1 has been summarized to avoid duplications with the
text. The reason why we examined peritumoral noncontrast-enhancing T2-hyperintense
areas was to evaluate if there were histological differences such as tumor
infiltration or increased vascularity in peritumoral regions between PCNSLs and GBs.
There have been several studies that showed increased rCBV on DSC-perfusion imaging
in peritumoral noncontrast-enhancing T2-hyperintense areas of GBs. This means that
such areas include not only vasogenic edema but also tumor cells infiltrating
surrounding brain parenchyma; however, our study did not reveal any significant
differences in GD model-based parameters for peritumoral noncontrast-enhancing
T2-hyperintense areas between PCNSLs and GBs. We assume that the high f2 values
observed in peritumoral noncontrast-enhancing T2-hyperintense areas compared to
enhancing areas in both types of tumor are likely to reflect perifocal vasogenic
edema rather than tumor infiltration outside the enhancing lesion. We have modified
the sentence in Materials and Methods and Discussion as follows.

Fig. 1. Region-of-interest (ROI). The ROIs were placed on postcontrast T1-weighted
images to include contrast enhancing lesions (A, C). The ROIs were also placed on
the non-contrast-enhancing T2-hyperintense areas surrounding the contrast-enhancing
area and the contralateral normal-appearing white matter (B, D).

Materials and Methods (Page 9, Line 14-21)

“The ROIs were also placed on the peritumoral non-contrast-enhancing T2-hyperintense
areas to evaluate whether there were differences in histological features including
tumor infiltration or increased vascularity in the peritumoral areas between PCNSLs
and GBs. In addition, the ROIs were placed on the contralateral normal-appearing
white matter. The ROIs for the peritumoral non-contrast-enhancing T2-hyperintense
areas and contralateral normal-appearing white matter were measured on the image
obtained with the b-value of 0 sec/mm2 image. The same ROIs were used for all DWI
analyses.”

Discussion (Page 23, Line 11- Page 24, Line 3)

“In the T2-hyperintense lesions without contrast enhancement, no significant
differences were observed between the PCNSL and GB groups for any parameters. There
have been several studies that showed increased rCBV on DSC-perfusion imaging in
peritumoral noncontrast-enhancing T2-hyperintense areas of GBs [36, 37]. The results
of these studies indicated that the peritumoral areas of GB include not only
vasogenic edema but also tumor cells infiltrating surrounding brain parenchyma;
however, our study did not reveal any significant differences in the GD model-based
parameters for peritumoral noncontrast-enhancing T2-hyperintense areas between
PCNSLs and GBs. The f2 values in the noncontrast-enhancing T2-hyperintense areas
were high in both types of tumor compared to those in the contrast-enhancing areas
and normal appearing white matter. We assume that the high f2 values in the
noncontrast-enhancing T2-hyperintense areas are likely to reflect mostly perifocal
vasogenic edema rather than tumor infiltration outside the enhancing lesion. Our
result is consistent with the previous DWI study in which ADC could not be used to
differentiate edema with infiltration of tumor cells from vasogenic edema in
high-grade gliomas and PCNSLs [38].”

New references

36. Law M, Cha S, Knopp EA, Johnson G, Arnett J, Litt AW. High-grade gliomas and
solitary metastases: differentiation by using perfusion and proton spectroscopic MR
imaging. Radiology. 2002;222(3):715-21. Epub 2002/02/28. doi:
10.1148/radiol.2223010558. PubMed PMID: 11867790.

37. Neska-Matuszewska M, Bladowska J, Sasiadek M, Zimny A. Differentiation of
glioblastoma multiforme, metastases and primary central nervous system lymphomas
using multiparametric perfusion and diffusion MR imaging of a tumor core and a
peritumoral zone-Searching for a practical approach. PLoS One. 2018;13(1):e0191341.
Epub 2018/01/18. doi: 10.1371/journal.pone.0191341. PubMed PMID: 29342201; PubMed
Central PMCID: PMCPMC5771619.

- Statistical Analysis

Page 9 – “The optimal cutoff point was determined by Youden's method.” – A literature
reference should be provided to support the method.

Response: The reference for Yuden’s method has been provided here.

28. Youden WJ. Index for rating diagnostic tests. Cancer. 1950;3(1):32-5. Epub
1950/01/01. doi: 10.1002/1097-0142(1950)3:1<32::aid-cncr2820030106>3.0.co;2-3.
PubMed PMID: 15405679.

Page 10 – “To determine whether the combination of multiple parameters for both the
GD model and the IVIM model increases the diagnostic performance, we first performed
a stepwise analysis to select appropriate parameters, and then we performed a
binomial logistic regression analysis to examine the combinations of the selected
parameters.” – Does “appropriate” means with the best diagnostic performances?
“Examine” in what way? This should be presented in a clearer way.

Response: We have provided more detailed explanations of the stepwise analysis in the
paragraph as follows.

Materials and Methods (Page 10, Line 11-20)

“To determine whether the combination of multiple parameters for both the GD model
and the IVIM model increases the diagnostic performance, we first performed a
stepwise analysis to select the explanatory variables for a multiple regression
model from a group of candidate variables by going through a series of automated
steps. A forward-selection rule was applied in which the analysis started with no
explanatory variables and then added variables, one by one, based on which variable
was the most statistically significant, until there were no remaining statistically
significant variables [30, 31]. We then performed a binomial logistic regression
analysis to examine the AUCs of the combinations of the selected parameters. Two
independent AUCs were compared using the method of Delong et al. [32].”

New references

30. Efroymson MA. Multiple regression analysis. In: Ralston A, Wilf HS, editors.
Mathematical methods for digital computers. New York: Wiley; 1960.

31. Smith G. Step away from stepwise. J Big Data. 2018;5(32). doi:
org/10.1186/s40537-018-0143-6.

• Results

- Comparisons of the parameters between the PCNSL and GB groups

Page 10 - 11 - “Figure 2 illustrates the results of our comparisons of the GD
model-derived parameters between the PCNSLs and GBs in the gadolinium-enhancing
lesions. The κ was significantly smaller in the PCNSL group (2.26 ± 1.00) than in
the GB group (3.62 ± 2.01, p=0.0004). The θ was not significantly different between
the groups. The f1 was significantly larger in the PCNSL group (0.542 ± 0.107) than
in the GB group (0.348 ± 0.132, p<0.0001). The f2 was significantly smaller in
the PCNSL group (0.372 ± 0.098) than in the GB group (0.508 ± 0.127, p<0.0001).
The f3 was also significantly smaller in the PCNSL group (0.086 ± 0.043) than in the
GB group (0.144 ± 0.062, p<0.0001). The detailed information for the parameters
is summarized in Table 1. Figure 3 provides a PCNSL case that showed a ring-like
enhancing mass lesion mimicking a GB. This lesion showed a low κ, a large f1, a
small f2, and a small f3, suggesting PCNSL. Figure 4 demonstrates a GB case that
showed a solid enhancing mass lesion. This lesion showed a small κ, a small f1,
moderate f2 and large f3, which are consistent with GB. In the T2-hyperintense areas
without contrast enhancement, no significant differences were found between the
PCNSL and GB groups for any of the GD model derived parameters.” - This paragraph
should be written in a more fluid way so the reader can better understand the ideas
that the authors chose to highlight.

Response: This paragraph has been rewritten and reorganized in a following order: 1)
the results for the gadolinium enhancing lesions, 2) the results for peritumoral
T2-hyperintense areas without contrast enhancement, 3) the results for normal
appearing white matter, 4) representative figures for PCNLS and GB. 

Results (Page 11, Line 6 – Page 12, Line 9)

“The detailed information for the parameters in the gadolinium enhancing lesion,
peritumoral T2-hyperintense areas without contrast enhancement, and normal appearing
white matter is summarized in Table 1. 

The results of our comparisons of the GD model-derived parameters between the PCNSLs
and GBs in the gadolinium-enhancing lesions are shown in Figure 2. In the
gadolinium-enhancing lesions, the κ was significantly smaller in the PCNSL group
(2.26 ± 1.00) than in the GB group (3.62 ± 2.01, p=0.0004), the f1 was significantly
larger in the PCNSL group (0.542 ± 0.107) than in the GB group (0.348 ± 0.132,
p<0.0001), the f2 was significantly smaller in the PCNSL group (0.372 ± 0.098)
than in the GB group (0.508 ± 0.127, p<0.0001), and the f3 was significantly
smaller in the PCNSL group (0.086 ± 0.043) than in the GB group (0.144 ± 0.062,
p<0.0001), while the θ was not significantly different between the groups. 

In the peritumoral T2-hyperintense areas without contrast enhancement, no significant
differences were found between the PCNSL and GB groups for any of the GD model
derived parameters. 

In the contralateral normal-appearing white matter, the f1 was significantly larger
in the PCNSL group (0.642 ± 0.047) than in the GB group (0.593 ± 0.044,
p<0.0001), the f2 was significantly smaller in the PCNSL group (0.316 ± 0.036)
than in the GB group (0.354 ± 0.038, p<0.0001), and the f3 was significantly
smaller in the PCNSL group (0.043 ± 0.026) than in the GB group (0.053 ± 0.019,
p=0.0105).

Figure 3 provides a PCNSL case that showed a ring-like enhancing mass lesion
mimicking a GB. This lesion showed a low κ, a large f1, a small f2, and a small f3,
suggesting PCNSL. Figure 4 demonstrates a GB case that showed a solid enhancing mass
lesion. This lesion showed a small κ, a small f1, moderate f2 and large f3, which
are consistent with GB.”

Page 14 – The results T2-hyperintense areas without contrast enhancement were
mentioned but no analysis was made to the NAWM results. If the results were obtained
these should be used to support the information shown about the GD model in the
manuscript.

Response: Thank you for pointing this out. We have added the statistical analyses on
the NAWM. Unexpectedly, there were significant differences in the f1, f2, and f3 of
NAWM between the PCNSLs and GB groups as follows. We have added the results and
discussion on these differences as follows. I found some errors in the values for
the NAWM in the GB group and have corrected them (the correction did not affect the
statistical results).

Results (Page 11, Line 21- Page 12, Line 4)

“In the contralateral normal-appearing white matter, the f1 was significantly larger
in the PCNSL group (0.642 ± 0.047) than in the GB group (0.593 ± 0.044,
p<0.0001), the f2 was significantly smaller in the PCNSL group (0.316 ± 0.036)
than in the GB group (0.354 ± 0.038, p<0.0001), and the f3 was significantly
smaller in the PCNSL group (0.043 ± 0.026) than in the GB group (0.053 ± 0.019,
p=0.0105).”

Discussion (Page 24, Line 4-9)

“In the normal-appearing white matter, the GB group showed larger f1, smaller f2, and
larger f3 than the PCNSL group although these differences were small. This was
unexpected, and the reasons for the differences remain unclear; however, since GBs
frequently show extensive infiltration into the surrounding brain tissue, which is a
fundamental feature of diffuse glioma, it is no wonder that the increased cell
density and perfusion were observed in the normal-appearing white matter.”

Page 14 – In the "all data" excel sheet for f parameter you use percentage and here
you show a fraction. Please make it uniform to clearly inform the reader.

Response: Thank you for pointing out. In the excel file named “all data”, the
f-values have been expressed as a fraction. 

Page 18 – “(…) although the difference was not significantly different from the
values for D (p=0.5276).” – This should be rephrased.

Response: This has been rephrased as follows.

Results (Page 19, Line 11-12)

“..., although the AUC for this combination was not significantly different from that
for D (p=0.5276).”

- Correlations of the model parameters

Page 18 – “The f1 had an almost perfect inverse correlation with D (all,

r = −0.976, p<0.0001; (…)” – According to figure 6, r = −0.9756, please confirm
the information and change where necessary.

Response: We have rounded the r-values to four decimal places as follows.

Results (Page 19, Line 18 - Page 20, Line 3)

“Figure 6 shows the correlations among the GD model-derived and IVIM model-derived
parameters in all tumors. The f1 had an almost perfect inverse correlation with D
(all, r = −0.9756, p<0.0001; PCNSL, r = −0.9558, p<0.0001; GB, r = −0.9699,
p<0.0001). The f2 had a very strong positive correlation with D (all, r=0.8865,
p<0.0001; PNCSL, r=0.9619, p<0.0001; GB, r=0.8273, p<0.0001). The f3 had a
very strong positive correlation with the f (all, r=0.8654, p<0.0001; PNCSL,
r=0.8317, p<0.0001; GB, r=0.8611, p<0.0001).”

Page 18 – Considering that the correlation between f2 and D parameters is 0.88657,
and it is mentioned because it is high compared to the other correlations, the
correlation between f1 and f2 should also be mentioned. The correlation between f1
and f2 is above 0.9155 and the meaning of this result can be of interest to be
discussed in the appropriate section of this work.

Response: Thank you for the suggestion. Yes, there was a very strong negative
correlation between f1 and f2. The following results and discussion regarding the
relationship between f1 and f2 have been added. The negative correlation between f1
and f2 was likely due to the complementary relationship between these two
parameters. Since the f3-values (PCNSL group, 0.086±0.043; GB group, 0.144±0.062)
were smaller compared to the f1- (PCNSL group, 0.542±0.107; GB group, 0.348±0.132)
and f2-values (PCNSL group, 0.372±0.098; GB group, 0.508±0.127), the increased f1
would result in the decreased f2, and vice versa.

Results (Page 20, Line 1-3)

“The f1 had an very strong negative correlation with the f2 (all, r = −0.9155,
p<0.0001; PCNSL, r = −0.9150, p<0.0001; GB, r = −0.8874, p<0.0001).”

Discussion (Page 23, Line 5-7)

“The negative correlation between f1 and f2 was likely due to the complementary
relationship between these two parameters. Since the f3-values were smaller compared
to the f1- and f2-values, the increase in f1 would result in the decrease in f2, and
vice versa.”

• Discussion

- The meaning of f3 and f parameters results and finding should be analysed in the
discussion. Why f and f3 parameters are higher in the PCNSL when compared to the GB
group of lesions?

Response: Thank you for your suggestion. Yes, this is an important point and we
should have mentioned this. The larger f3 and f values in the GB group compared to
those in the PCNSL group can be attributed to the difference in vascularity of these
tumors. Pathologically, neovascularization is a key feature of GB while it is not
prominent in PCNSL. Our results for f3 and f are consistent with those from previous
studies using dynamic susceptibility contrast perfusion-weighted MR imaging and
arterial spin labeling imaging. We have added the discussions on the meaning of f3
and f as follows.

“The GB group showed larger f3 and f compared to the PCNSL group. This may be
attributed to the difference in vascularity of these tumors. Pathologically,
neovascularization is a key feature of GB while it is not prominent in PCNSL [33,
34]. Our results are consistent with those from previous studies using dynamic
susceptibility contrast perfusion-weighted imaging and arterial spin labeling
imaging [9, 35].”

New references

33. Hardee ME, Zagzag D. Mechanisms of glioma-associated neovascularization. Am J
Pathol. 2012;181(4):1126-41. Epub 2012/08/04. doi: 10.1016/j.ajpath.2012.06.030.
PubMed PMID: 22858156; PubMed Central PMCID: PMCPMC3463636.

34. Cha S, Knopp EA, Johnson G, Wetzel SG, Litt AW, Zagzag D. Intracranial mass
lesions: dynamic contrast-enhanced susceptibility-weighted echo-planar perfusion MR
imaging. Radiology. 2002;223(1):11-29. Epub 2002/04/04. doi:
10.1148/radiol.2231010594. PubMed PMID: 11930044.

35. Hartmann M, Heiland S, Harting I, Tronnier VM, Sommer C, Ludwig R, et al.
Distinguishing of primary cerebral lymphoma from high-grade glioma with
perfusion-weighted magnetic resonance imaging. Neurosci Lett. 2003;338(2):119-22.
Epub 2003/02/05. doi: 10.1016/s0304-3940(02)01367-8. PubMed PMID: 12566167.

- Another important result, which should be discussed, is that with the GD model
parameters (κ, f1, and f3) it was possible to obtain a higher AUC when compared to
ADC’s AUC. Can the author provide a justification or further analysis?

Response: Thank you very much. As pointed out, the AUC of this combination (0.909)
was higher than that of ADC (0.879). Although there was no statistically significant
difference between them (p=0.2152), this tendency could be meaningful and important.
Whether the combination of parameters of the GD model has an additive value should
be evaluated in a larger population. We have added the following sentences in
Results and Discussion.

Results (Page 19, Line 6-8)

“The AUC of this combination (0.909) was higher than that of ADC (0.879); however,
there was no statistically significant difference between them (p=0.2152).”

Discussion (Page 22, Line 16-19)

“The AUC of this combination tended to be higher than that of ADC although there was
no significant difference. Whether the combination of parameters of the GD model has
an additive value should be evaluated in a larger population, since we did not
observe statistical significance in all of our comparisons.”

Page 20 – “The reason for the slightly higher AUC observed with the ADC could be the
effect of perfusion on ADC measurements.” - Can the effect of perfusion on ADC
explain the higher AUC observed? If that is the case, in what way can this be
explained?

Response: I am sorry that our discussion lacked explanations for this result. The ADC
is a diffusion coefficient but is influenced by tissue perfusion (that's why this is
called “apparent” diffusion coefficient). In hyperperfused tissues, ADC will be
affected by perfusion and overestimated compared to true diffusion coefficient D,
while the perfusion effect will be excluded in the calculation of D. In
hypervascular tumors such as GBs, the ADC would be larger than D. On the other hand,
in hypovascular tumors such as PCNSLs, this difference between ADC and D should be
smaller. This means that the difference between ADC and D would be larger in GBs
than in PCNSLs. Therefore, ADC could show higher diagnostic performance in the
discrimination of these two tumors than D. This paragraph has been modified and
added more explanations as follows.

Discussion (Page 22, Line 3-14)

“The reason for the slightly higher AUC observed with the ADC could be the effect of
perfusion on ADC measurements. In hyperperfused tissues, ADC will be affected by the
perfusion effect and overestimated compared to D; however, since both f1 and D are
parameters without a perfusion effect in theory, an overestimation caused by
perfusion should not be observed in these values. Therefoere, in hypervascular
tumors such as GBs, the ADC should be larger than D. On the other hand, in
hypovascular tumors such as PCNSLs, this difference between ADC and D should
smaller. This means that the difference between ADC and D would be larger in GBs
than in PCNSLs. Therefore, ADC could show higher diagnostic performance in the
discrimination of these two tumors than D. In fact, the difference between the ADC
and D values was greater in the GBs (0.100 × 10−3 mm2/sec) than in the PCNSLs
(0.078 × 10−3 mm2/sec), which was most likely due to the higher perfusion effect on
the ADC in GBs than in PCNSLs.”

Page 21 – “We found the correlations between the GD model-derived and IVIM-derived
parameters.” – This sentence is vague or incomplete please consider revising it.

Response: This sentence has been revised to be more specific as follows.

Results (Page 22, Line 21-22)

“We found the correlations between the GD model-derived and IVIM-derived parameters,
particularly between the f1 and D, the f2 and D, and the f3 and f.”

Page 21 – “The almost perfect correlation observed between the f1 and D may indicate
that these two parameters contain virtually identical information.” – It would be
important to mention that it is a negative correlation.

Response: The term “negative” has been added as requested.

Discussion (Page 22, Line 1)

“The almost perfect negative correlation observed between the f1 and D may indicate
that these two parameters contain virtually identical information.”

Page 21 – “The positive correlation between f2 and D suggests the opposite, and the
increased extracellular space like that taken up by microscopic necrosis might
result in the higher f2.” – The “opposite” to what?

Response: This wording was not appropriate, and thus the term “opposite” has been
deleted as follows.

Discussion (Page 23, Line 2-3)

“The positive correlation between f2 and D suggests that the increased extracellular
space like that taken up by microscopic necrosis might result in the higher f2.”

Page 21 – “Although the GD-derived and IVIM-derived parameters provide similar
information, the strength of the GD model-derived parameters is that the values are
expressed as a fraction or percentage, which allows us to characterize

tumors from histological viewpoint.” - “f “ is an IVIM-derived parameter and it is
also expressed in percentage or fraction.

Response: Yes, as you pointed out, the IVIM-f is expressed in percentage or fraction.
But the IVIM method cannot express the fractions or percentage of intracellular and
extracellular-extravascular spaces. Therefore, we think IVIM is not a perfect method
to characterize tumors from histological viewpoint.

Page 21 – “The high f2 values in both types of tumor are likely to reflect mostly
perifocal vasogenic edema rather than tumor infiltration outside the enhancing
lesion” – “high f2 values” relative to what? The sentence would be more specific if
it mentioned that it is referring to T2-hyperintense lesions. This idea would
benefit from an addition of a literature reference.

Response: The f2 values in the noncontrast-enhancing T2-hyperintense areas were
higher in both types of tumor compared to those in the contrast-enhancing areas and
normal appearing white matter. The sentence has been revised as follows. This is a
new finding and we could not find an appropriate reference related to this result. A
new reference has been added.

Discussion (Page 23, Line 19 – Page 24, Line 4)

“The f2 values in the noncontrast-enhancing T2-hyperintense areas were higher in both
types of tumor compared to those in the contrast-enhancing areas and normal
appearing white matter. We assume that the high f2 values in the
noncontrast-enhancing T2-hyperintense areas are likely to reflect mostly perifocal
vasogenic edema rather than tumor infiltration outside the enhancing lesion. Our
result is consistent with the previous DWI study in which ADC could not be used to
differentiate edema with infiltration of tumor cells from vasogenic edema in
high-grade gliomas and PCNSLs [38].

New refference

38. Server A, Kulle B, Maehlen J, Josefsen R, Schellhorn T, Kumar T, et al.
Quantitative apparent diffusion coefficients in the characterization of brain tumors
and associated peritumoral edema. Acta Radiol. 2009;50(6):682-9. Epub 2009/05/19.
doi: 10.1080/02841850902933123. PubMed PMID: 19449234.

Page 22 – Another limitation is the fact that only one person performed the ROI
placement. The fact that the highest b-value in use was 1000s/mm2, should be better
justified since in the brain higher b-values are usually used.

Response: The limitation for the ROI placement has been added. In a study of prostate
cancers, Oshio et al. used the similar DWI parameters to ours and highest b-value of
1000s/mm2, and reported that the good fitting accuracy was observed in both cancer
(R2=0.99226) and normal tissue (R2=0.99842). Their results indicated that DWI with
the highest b-value of 1000 s/mm2 can be used for GD model analysis. We have
modified the paragraph to justify the use of this b-value as follows.

Discussion (Page 24, Line 12-14)

“The only one person performed the ROI placements on a single slice, and not whole
tumor volume was evaluated.”

Discussion (Page 24, Line 17- Page 25, Line 5)

“In addition, the selection of b-values has not yet been optimized. Prior studies of
the GD model used the maximum b-values ranging from 1000 to 3000 sec/mm2 [18,
22-24]. In a study of prostate cancers, Oshio et al. used the similar DWI parameters
to ours and the highest b-value of 1000 s/mm2, and reported that the good fitting
accuracy was observed in both cancerous tissues (R2=0.99226) and normal tissues
(R2=0.99842) [22]. Their result indicated that DWI with the highest b-value of 1000
s/mm2 can be used for GD model analyses; however, since it was reported that the
non-monoexponential diffusion-related signal decay generally becomes more obvious
over more extended b-value ranges, the maximum b value of 1000 sec/mm2 used in the
present study might be lower than the optimal value. The optimal b-values and
numbers should be elucidated in future studies.”

- Figures

Figure 1. It would be interesting to have the non-contrast-enhancing T2 image where
the ROIs were also placed.

Response: Another case (new Figure 1 C, D) which well illustrates
noncontrast-enhancing T2-hyperintense areas has been added.

Figure 5 The y axis title is missing

Response: Thank you very much! This has been corrected.

Figure 6 Units should be included, when appropriate.

Response: The units have been added for D and D*. The f1, f2, f3 and f are expressed
as fractions and no units are necessary for them.

- Tables

Table 1 It would be easier to interpret the information shown in the table if
p-values were also presented.

Response: The p-values have been added in Table 1 as suggested.

Table 2 In this table the reader is first exposed to the combination of parameters
that were used to estimate diagnostic performances, additionally to the estimation
of the diagnostic performances with the individual parameters. Why these specific
combinations of parameters were considered? The justification should be clearly
stated in the text.

In the legend “D, true diffusion coefficient; D*, ç; f, perfusion fraction” the
meaning of D* is missing.

Response: The stepwise analysis selected the combination of κ, f1, and f3. This is
already stated in the text as follows. The meaning of D* (pseudo-diffusion
coefficient) has been corrected.

Results (Page 19, Line 1-4)

“In the combined-parameters analysis, the stepwise procedure selected κ, f1, and f3
for the GD model, and the D and f for the IVIM model. The combination of κ, f1, and
f3 revealed excellent diagnostic performance with the AUC of 0.909, sensitivity of
84.2%, and specificity of 88.9%.”

- Supporting information

Units should be added to the “all data” table where appropriate.

Response: The units have been added in all data table.

----Statement:

The topic is interesting but there are some parts of the study which the description
should be improved so the reader can better understand it.

The major strengths of the article are: the application of a new diffusion model to
brain tumours, which as far as I know it has not been done yet; the inclusion of an
analysis of combined parameters and the evaluation of its performance; the inclusion
of healthy tissue results, but these results were not compared to the results obtain
for tumours which is a weakness. Consequently, the relevant weaknesses of the work
are: the need for a deeper analysis of the obtained results; there are some methods
and procedures that should be better described and justified; the use of b=1000
s/mm2 as the highest b-value is unusual in a brain diffusion studies, and this
should be further justified.

Response: We sincerely appreciate the reviewer’s positive comments. We believe that
we have corrected all the weaknesses of the present study which the reviewer pointed
out as seen above.

Note: “Have the authors made all data underlying the findings in their manuscript
fully available?” My answer was “no” because I only had access to the summary
statistics, and not to the data points behind the statistics.

Response: In the supplementary data named “all data”, all raw data of measurements
are available. We forgot to include raw data for T2-hyperintense areas and NAWM in
the previous file and have added them in the new file.

Reviewer #2: This study examined gamma distribution model of diffusion MRI and this
model is useful to differentiate malignant lymphoma and glioblastoma.

This is well written paper and there are some minor points to revise.

In table 1, clarify the statistical significant differences between PCNSL and GB.

Response: We have added all p-values in Table 1 as suggested.

In figure 4, the k and f2 values at lateral peritumor area (lateral side) are high
compared to contrast-enhanced tumor area.

Response: Yes, the peritumoral T2-hyperintense area without contrast enhancement
showed a large κ and a large f2 compared to the contrast-enhancing area. These
findings are consistent with the results of the present study (Table 1). We have
added the descriptions about the peritumoral T2-hyperintense areas in Fig. 3 and 4.
Thank you very much for your suggestion.

Fig. 3. The peritumoral T2-hyperintense area without contrast enhancement shows a
large κ (8.18, D), a small θ (0.46×10−6 mm2/sec, E), a small f1 (0.139, F), a large
f2 (0.772, G), and a small f3 (0.090, H).

Fig. 4. The peritumoral T2-hyperintense area without contrast enhancement shows a
large κ (3.94, D), a small θ (0.75×10−6 mm2/sec, E), a small f1 (0.308, F), a large
f2 (0.597, G), and a small f3 (0.095, H).

Discussion

Limitation. The authors evaluate the only one slice of tumor, not whole tumor volume.
Add this point in the limitation.

Response: We agree with this point. This has been added in the limitation as
follows.

Discussion (Page 24, Line 12-14)

“The only one person performed the ROI placements on a single slice, and not whole
tumor volume was evaluated.”

to Reviewers.docx
---

## [Decision Letter · Decision Letter 1]

5 Oct 2020

PONE-D-20-15410R1

Gamma distribution model of diffusion MRI for the differentiation of primary central
nerve system lymphomas and glioblastomas

PLOS ONE

Dear Dr. Togao,

Thank you for submitting your manuscript to PLOS ONE. After careful consideration, we
feel that it has merit but does not fully meet PLOS ONE’s publication criteria as it
currently stands. Therefore, we invite you to submit a revised version of the
manuscript that addresses the points raised during the review process.

There are just a few relatively minor issues to be addressed as pointed out by
Reviewer 1.

Please submit your revised manuscript by Nov 19 2020 11:59PM. If you will need more
time than this to complete your revisions, please reply to this message or contact
the journal office at plosone@plos.org. When
you're ready to submit your revision, log on to https://www.editorialmanager.com/pone/ and select the 'Submissions
Needing Revision' folder to locate your manuscript file.

If you would like to make changes to your financial disclosure, please include your
updated statement in your cover letter. Guidelines for resubmitting your figure
files are available below the reviewer comments at the end of this letter.

We look forward to receiving your revised manuscript.

Kind regards,

Niels Bergsland

Academic Editor

PLOS ONE

Reviewers' comments:

Reviewer's Responses to Questions

**Comments to the Author**

1. If the authors have adequately addressed your comments raised in a previous round
of review and you feel that this manuscript is now acceptable for publication, you
may indicate that here to bypass the “Comments to the Author” section, enter your
conflict of interest statement in the “Confidential to Editor” section, and submit
your "Accept" recommendation.

Reviewer #1: (No Response)

Reviewer #2: All comments have been addressed

2. Is the manuscript technically sound, and do the data
support the conclusions?

Reviewer #1: (No Response)

Reviewer #2: Yes

3. Has the statistical analysis been performed
appropriately and rigorously? 

Reviewer #1: (No Response)

Reviewer #2: Yes

4. Have the authors made all data underlying the
findings in their manuscript fully available?

Reviewer #1: (No Response)

Reviewer #2: Yes

5. Is the manuscript presented in an intelligible
fashion and written in standard English?

Reviewer #1: (No Response)

Reviewer #2: Yes

6. Review Comments to the Author

Reviewer #1: The authors have addressed the topics indicated in the first revision
and the manuscript has been significantly improved. Here, the authors can find some
small details that still need to be considered.

• Results (Page 19, Line 11-12)

“..., although the AUC for this combination was not significantly different from that
for D (p=0.5276).”

"This combination increased the diagnostic performance of f (p=0.0077), although the
AUC for this combination was not significantly different from that for D
(p=0.5276)."

- The language used in the phrases “(…) significantly different from that for
(…)”through all the text should be rewritten in a clearer way in order to sound.

• Discussion (Page 23, Line 5-7)

“The negative correlation between f1 and f2 was likely due to the complementary
relationship between these two parameters. Since the f3-values were smaller compared
to the f1- and f2-values, the increase in f1 would result in the decrease in f2, and
vice versa.”

- An explanation for this relation should be put forward taking into account the
meaning of the parameters.

• “Page 21

“Although the GD-derived and IVIM-derived parameters provide similar information, the
strength of the GD model-derived parameters is that the values are expressed as a
fraction or percentage, which allows us to characterize tumours from histological
viewpoint.” - “f “ is an IVIM-derived parameter and it is also expressed in
percentage or fraction.

Response: Yes, as you pointed out, the IVIM-f is expressed in percentage or fraction.
But the IVIM method cannot express the fractions or percentage of intracellular and
extracellular-extravascular spaces. Therefore, we think IVIM is not a perfect method
to characterize tumours from histological viewpoint.”

- In the way this sentence is presented, the reader may misunderstand the information
that you are providing. You are stating that one strong point of the GD model
parameters, when compared to IVIM parameters, is to be presented in percentage or
fraction and that this is the reason why it allows the characterization of tumours.
f is also expressed as fraction or percentage and it shows problems in this task. In
the way that the sentence is constructed the reader may think that only GD
parameters are expressed in fraction or percentage. Also, it is important to refer
in what way can this characteristic contribute to the characterization of tumours'
histology.

• Figure 1

- The name of the lesions should be included. Also, the lesions/ROIs in the images
should be identified for example with numbers or letters, and that should be
referenced and related in the legend of the figure.

Reviewer #2: The paper is revised as reviewer's comments and it is acceptable in this
version.

This is very useful information for brain tumor imaging.

7. PLOS authors have the option to publish the peer
review history of their article (what does this mean?). If published, this will
include your full peer review and any attached files.

If you choose “no”, your identity will remain anonymous but your review may still be
made public.

**Do you want your identity to be public for this peer review?** For
information about this choice, including consent withdrawal, please see our
Privacy Policy.

Reviewer #1: No

Reviewer #2: **Yes: **Yoshiyuki Watanabe

---

## [Author Response · Author response to Decision Letter 1]

13 Nov 2020

Reviewer #1: The authors have addressed the topics indicated in the first revision
and the manuscript has been significantly improved. Here, the authors can find some
small details that still need to be considered.

• Results (Page 19, Line 11-12)

“..., although the AUC for this combination was not significantly different from that
for D (p=0.5276).”

"This combination increased the diagnostic performance of f (p=0.0077), although the
AUC for this combination was not significantly different from that for D
(p=0.5276)."

- The language used in the phrases “(…) significantly different from that for
(…)”through all the text should be rewritten in a clearer way in order to sound.

Response: First, we would like to thank you for your thorough reading and
constructive criticism which greatly enhanced the quality of the paper. According to
your comments, we have carefully revised the paper again. The sentences have been
rewritten as follows.

“This combination increased the diagnostic performance of κ (p=0.0016), and f3
(p=0.0075), although it did not improve the performance of f1 (p=0.1950).” (Page 19,
Line 4-6)

“This combination improved the diagnostic performance of f (p=0.0077), although it
did not improve the performance of D (p=0.5276).” (Page 19, Line 9-10)

• Discussion (Page 23, Line 5-7)

“The negative correlation between f1 and f2 was likely due to the complementary
relationship between these two parameters. Since the f3-values were smaller compared
to the f1- and f2-values, the increase in f1 would result in the decrease in f2, and
vice versa.”

- An explanation for this relation should be put forward taking into account the
meaning of the parameters.

Response: The relationship between f1 and f2 has been explained considering the
meaning of the parameters as follows.

“The negative correlation between f1 and f2 was likely due to the complementary
relationship between these two parameters. In general, intravascular space (≒ f3) is
smaller compared to intracellular (≒ f1) and extracellular extravascular space (≒
f2). In fact, the f3-values were much smaller than the f1- and f2-values in both
PCNSLs and GBs in the present study. Therefore, the increase in f1 would result in
the decrease in f2, and vice versa.” (Page 23, Line 4-9)

• “Page 21

“Although the GD-derived and IVIM-derived parameters provide similar information, the
strength of the GD model-derived parameters is that the values are expressed as a
fraction or percentage, which allows us to characterize tumours from histological
viewpoint.” - “f “ is an IVIM-derived parameter and it is also expressed in
percentage or fraction.

Response: Yes, as you pointed out, the IVIM-f is expressed in percentage or fraction.
But the IVIM method cannot express the fractions or percentage of intracellular and
extracellular-extravascular spaces. Therefore, we think IVIM is not a perfect method
to characterize tumours from histological viewpoint.”

- In the way this sentence is presented, the reader may misunderstand the information
that you are providing. You are stating that one strong point of the GD model
parameters, when compared to IVIM parameters, is to be presented in percentage or
fraction and that this is the reason why it allows the characterization of tumours.
f is also expressed as fraction or percentage and it shows problems in this task. In
the way that the sentence is constructed the reader may think that only GD
parameters are expressed in fraction or percentage. Also, it is important to refer
in what way can this characteristic contribute to the characterization of tumours'
histology.

Response: Thank you very much for the comments. We have modified the sentences as
follows according to your suggestion. We think that obtaining all fraction values
(f1, f2, f3) would help to characterize tumors from histological viewpoint since f1,
f2, and f3 should reflect cell density, interstitial space, and vascularity,
respectively. 

“Although the GD-derived and IVIM-derived parameters provide similar information, the
strength of the GD model-derived parameters is that all fraction values (f1, f2, f3)
are expressed as fractions or percentages, which allows us to well characterize
tumors from histological viewpoint. The IVIM-derived f-value is also expressed in a
percentage or fraction; however, the IVIM analysis is not able to provide the
fraction values for intracellular and extracellular-extravascular spaces. In this
sense, the IVIM method is not a perfect method for the histological characterization
of tumors.”(Page 23, Line 9-15)

• Figure 1

- The name of the lesions should be included. Also, the lesions/ROIs in the images
should be identified for example with numbers or letters, and that should be
referenced and related in the legend of the figure.

Response: Thank you for your suggestions. The names of the lesions have been included
in the legend. The ROI numbers (#1, contrast enhancing areal; #2 T2-hyperintense
area; #3, normal appearing white matter) have been added in the Figures and have
been related in the legends as follows.

Fig. 1. Regions-of-interest (ROIs). Figures 1 A and B show a GB with ring
enhancement, and Figures C and D show a PCNSL with solid enhancement. The ROIs were
placed on postcontrast T1-weighted images to include contrast enhancing lesions (A,
C, area #1). The ROIs were also placed on the non-contrast-enhancing T2-hyperintense
areas surrounding the contrast-enhancing area (area #2) and the contralateral
normal-appearing white matter (B, D, area #3).

Reviewer #2: The paper is revised as reviewer's comments and it is acceptable in this
version.

This is very useful information for brain tumor imaging.

Response: We appreciate the reviewer’s comments.

---

## [Decision Letter · Decision Letter 2]

30 Nov 2020

Gamma distribution model of diffusion MRI for the differentiation of primary central
nerve system lymphomas and glioblastomas

PONE-D-20-15410R2

Dear Dr. Togao,

We’re pleased to inform you that your manuscript has been judged scientifically
suitable for publication and will be formally accepted for publication once it meets
all outstanding technical requirements.

Kind regards,

Niels Bergsland

Academic Editor

PLOS ONE

Additional Editor Comments (optional):

Reviewers' comments:

Reviewer's Responses to Questions

**Comments to the Author**

1. If the authors have adequately addressed your comments raised in a previous round
of review and you feel that this manuscript is now acceptable for publication, you
may indicate that here to bypass the “Comments to the Author” section, enter your
conflict of interest statement in the “Confidential to Editor” section, and submit
your "Accept" recommendation.

Reviewer #1: All comments have been addressed

2. Is the manuscript technically sound, and do the data
support the conclusions?

Reviewer #1: Yes

3. Has the statistical analysis been performed
appropriately and rigorously? 

Reviewer #1: (No Response)

4. Have the authors made all data underlying the
findings in their manuscript fully available?

Reviewer #1: (No Response)

5. Is the manuscript presented in an intelligible
fashion and written in standard English?

Reviewer #1: Yes

6. Review Comments to the Author

Reviewer #1: The authors addressed the topics indicated in the previous review. The
article provides an understanding of an important topic by applying a method that
can bring more in-depth knowledge to this field of study.

Note that in Figure 1, the numbers that identify the ROIs are covering the relevant
structures, which can raise doubts among readers. Please consider a solution, as for
example the use of different colours to differentiate the ROIs instead, or a
different positioning of the numbers.

7. PLOS authors have the option to publish the peer
review history of their article (what does this mean?). If published, this will
include your full peer review and any attached files.

If you choose “no”, your identity will remain anonymous but your review may still be
made public.

**Do you want your identity to be public for this peer review?** For
information about this choice, including consent withdrawal, please see our
Privacy Policy.

Reviewer #1: **Yes: **Filipa Borlinhas

---

## [Editor Report · Acceptance letter]

3 Dec 2020

PONE-D-20-15410R2 

Gamma distribution model of diffusion MRI for the differentiation of primary central
nerve system lymphomas and glioblastomas 

Dear Dr. Togao:

I'm pleased to inform you that your manuscript has been deemed suitable for
publication in PLOS ONE. Congratulations! Your manuscript is now with our production
department. 

Kind regards, 

on behalf of

Dr. Niels Bergsland 

Academic Editor

PLOS ONE